

# Earth's radiation belts ions: Patterns of the spatial-energy structure and its solar-cyclic variations

**Alexander S. Kovtyukh**

Skobeltsyn Institute of Nuclear Physics, Moscow State University, Moscow, 119234, Russia;
kovtyukhas@mail.ru

**Abstract** Spatial-energy distributions of the stationary fluxes of protons, helium ions and ions of carbon-nitrogen-oxygen (CNO) group, with energy from $E \sim 100$ keV to 200 MeV, in the Earth's radiation belts (ERB), at $L \sim 1$–8, are considered here by the data of the satellites for 1961–2017. It is find that the results of these measurements line up in the space $\{E, L\}$ by some regular patterns. Solar-cyclic (11-year) variations in the distributions of protons, helium ions and CNO group ions fluxes in the ERB are studied. In the inner regions of the ERB the ions fluxes decrease with increasing solar activity. It is find, the solar-cyclic variations of fluxes for ions with $Z \geq 2$ are much greater than for protons and increase with increasing an atomic number $Z$ of the ions. The possible physical mechanisms leading to formation of this spatial-energy structure and to the solar-cyclic variations of the ERB ion fluxes are discussed.

**Keywords**. Magnetospheric physics (energetic particles, trapped). Radiation belts.



# 1 Introduction

The Earth's radiation belts (ERB) consist of charged particles with energy from $E \sim 100$ keV to
several hundreds of megaelectronvolt (MeV). These particles are trapped by the geomagnetic field
at altitudes from $\sim 200$ kilometers to $\sim 50$–70 thousands kilometers. The ERB is consisted mainly
from electrons and protons. In the ERB there are also ions of helium, oxygen, and other elements
with the atomic number $Z \geq 2$, where $Z$ is the charge of the atomic nucleus with respect to the
charge of the proton. During geomagnetic disturbances a fluxes of ions and its distributions are
varied. These fluxes depend also on the phase of the solar cycle, conditions in the interplanetary
space, and other factors.
Particles with different energy $E$ and pitch angles $\alpha$ ($\alpha$ is the angle between a local vector of the
magnetic field and vector of a particle velocity), which injected into some point of the geomagnetic
trap, are drifted with conserving the adiabatic invariants ($\mu$, $K$, $\Phi$) and populate a narrow layer
surrounding the Earth (Alfvén and Fälthammar, 1963; Northrop, 1963). This layer call the drift
shell. Therefore, experimental data on the ERB do most simply represented in coordinates $\{L, B\}$,
where $L$ is parameter of a drift shell and $B$ is a local induction of the magnetic field (McIlwain,
1961). For the dipole magnetic field $L$ is a distance, in the equatorial plane, from the given
magnetic field line to the center of the dipole (in the Earth's radii $R_E$).
The stationary fluxes $J$ of the ERB particles with given energy and pitch angle $\alpha$ are decreased
usually when the point of observation is shifted from the equatorial plane to a higher latitudes
along certain magnetic field line (if we exclude the peripheral regions of the geomagnetic trap,
where the drift shells of the captured particles are split and branched). This dependence of the
particle fluxes is described by the functions $J(B/B_0)$, were $B$ and $B_0$ are values of the magnetic field
at the point of observation and in the equatorial plane on the same magnetic field line.
Outer and inner regions of the ERB maintained in the dynamic equilibrium with the
environment by the different mechanisms (see review Kovtyukh, 2018).
The outer belt ($L > 3.5$) of ions is formed mainly by the mechanisms of the radial diffusion of
these ions to the Earth under the action of fluctuations of an electric and magnetic fields resonating
with a drift periods of these ions. This transport accompanied by the betatron acceleration of ions
and by the ionization losses of the ions in result of their interactions with the plasmasphere and
with residual atmosphere.
The inner belt ($L < 2.5$) of protons with $E > 10$ MeV is formed mainly as a result of decay of
neutrons knocked from the nuclei of the atmospheric atoms by the Galactic Cosmic Rays (GCR).
For protons with $E < 10$ MeV this mechanism (CRAND) is supplemented by the radial diffusion
of particles from the outer to the inner belt. The inner belt of ions with $Z > 4$ was formed mainly
from the ions of the Anomalous component of Cosmic Rays (ACR).
In the intermediate region ($2.5 < L < 3.5$) is operated also the mechanism of a capture of the
ions from the Solar Cosmic Rays (SCR) during strong magnetic storms (see, e.g., Selesnick et al.,
56  2014).

Thus, the main mechanisms of formation of the ERB, sources and losses of the ions are known.
However, for the comprehensive verification of the physical models and to identification of the
mathematical models parameters it is necessary sufficiently complete and reliable empirical
models of the ERB for each of ion components. It is necessary also for ensuring the safety of space
flights.
These models can be created only on a basis of the experimental data are obtained over many
years and decades. Such models (see, e.g., Ginet et al., 2013) were created for protons (AP8/AP9).
These models are widely used in the space research. However, measurements of fluxes of the ions
with $Z \geq 2$ are represented a difficult technical problem due to a small fluxes of these ions and
high background fluxes of protons and electrons. The empirical and semi-empirical models of the
ERB developed for ions with $Z \geq 2$ are applicable only to very limited regions of the space $\{E, L\}$.





There are problems connected with limited and incomplete information on the fluxes of ions
with $Z \geq 2$ in the ERB, especially in the energy range from tens to hundreds of megaelectronvolt.
One of the main problem of this work is to consider the possibility to create sufficiently complete
and reliable empirical models of the ERB for these ions based on currently available world
experimental data.
In the following sections are considered the spatial-energy structure of the ERB in the spaces
$\{E, L\}$ for protons, helium ions and ions of the CNO group on the experimental data (Sect. 2),
possible physical mechanisms of formation of these structures and its solar-cyclic variations are
discussed (Sect. 3) and the main conclusions of this work are given (Sect. 4).

## 2  Spatial-energy distributions of the ion fluxes near the equatorial plane

There can be trapped on the drift shells only ions with energy less than some maximum values
determining by the Alfvén's criterion: $\rho_i(L, E, M_i, Q_i) << R_c(L)$, where $\rho_i$ is the gyroradii of ions,
and $R_c$ is the radius of curvature of the magnetic field near the equatorial plane ($M_i$ and $Q_i$ are mass
and charge of ions with respect to the corresponding values for protons). According to this
criterion and the theory of stochastic motion of particles, the geomagnetic trap can capture and
durably hold only ions with $E$ (MeV) $< 2000 \times (Q_i^2/M_i)\, L^{-4}$ (Ilyin et al., 1984). The green line is
present this boundary on Figs. 1–6.
When comparing the data of various experiments in the ERB, the question arises about the
compatibility of these results with each other and the reasons for their discrepancies. More or less
significant discrepancies in the results of the satellites can be connected with the differences in the
trajectories of the satellites; in the construction of the instruments and their angular characteristics;
in the energy ranges and sets of the energy channels. For the stationary ERB, these discrepancies
can also be associated with differences in the general state of the Solar, heliosphere and
magnetosphere of the Earth at the different measurements periods. These factors influence on the
fluxes of ions with $Z \geq 2$ in the ERB more significantly than on the proton fluxes (see, e.g.,
Kovtyukh, 2018).
This section used experimental data of various satellites, which were obtained for quiet periods
(Kp < 2) and near the equatorial plane of the ERB for ions with equatorial pitch angles $\alpha_0 \approx 90^o$.
The preference was given to the averaged results of these satellites for quiet periods. All values of
differential fluxes reduced to one dimension. In the regions of $E$ and $L$ shells where these data were
obtained the ion fluxes do not distorted by the background of other particles.
In many important experiments, the instruments did not allow separate fluxes of ions by charge
of ions. For ions of the CNO group the separation by mass also are not performing usually. For
heavier ions, for example for Fe ions, we have very smaller such data. Therefore, this work
presents only helium ions (without separating them by charge) and ions of CNO group (without
separating them by mass and charge).
To solve the problems considered here, it is important to choose the form of representation
(space of variables), in which the results of different experiments can be conformed to each other
naturally. For such representation of the distributions of the ion fluxes have been chosen the space
$\{E. L\}$. Such representation is possible to organize of a fragmentary experimental data obtained in
different ranges of $E$ and $L$ most effectively.
Figures 1–6 presented here the spatial-energy distributions of the fluxes of protons, helium ions,
and ions of the CNO group near the equatorial plane. These figures are paired: odd figures refer to
periods near the minima, and even figures refer to periods near the solar activity maxima. The
values $E$ and $L$ in these figures are presented in logarithmic scales. Experimental points on these
figures connected by lines of the equal intensity of ion fluxes (iso-lines); the decimal logarithms of
the fluxes $J$ are shown near each iso-lines. The ion fluxes $J$ have a dimension $(cm^2\ s\ ster\ MeV/n)^{-1}$
and are corresponded to the energies $E$ (MeV/n) and an equatorial pitch angle of $\alpha_0$ is $\sim 90^o$.



Such representations of the experimental data are not only visual, but also are very convenient
and rather universal. Obviously, Figs. 1–6 actually show both radial profiles of the fluxes of ions
for a given energy (if one see along the abscissa axis) and ion energy spectra for a given $L$ shell (if
one see along the ordinate axis).
In this place, it is need to say a few words about the method of constructing these figures. The
points in Figs. 1–6 are obtained from the dependences $J(L)$ for a given ions with a certain energy
(the average energy for each channel) and with an equatorial pitch angle close to $90^{\mathrm{o}}$. Unlike a
distributions of electrons, as well as ion distributions connected with magnetic disturbances, the
dependences $J(L)$ for the ERB ions with $\alpha_0 \sim 90^{\mathrm{o}}$ usually have only one maximum in a quiet time.
As a result, for each experiment 1 or 2 points were obtained (on the outer and inner edges of this
profile) with certain values of $E$ and $L$ for a given level of stationary ion fluxes. Sometimes,
especially for low levels of fluxes, only one point was obtained: in these cases, the radial profile of
the ion fluxes was cutoff at small $L$ by a significant background of other particles. In these cases,
no interpolations and extrapolations of the radial profiles of ion fluxes does performing here.
Each iso-line shown in these figures was built separately, for the corresponding set of
experimental points (icons); after that this iso-line was transferred (along with the icons) to the
corresponding figure. Thus, in the region abundantly populated by such icons (for protons with $E >$
1 MeV at $L > 2$) they are mixing in Figs. 1–2. In the cases of a large distances between neighbor
points, the corresponding segments of the iso-lines are shown in dotted arcs in these figures.
The radial profiles of the differential fluxes $J(L)$ shown on uniform presentation for particles of
different energies are intersect with each other in those regions where the energy spectra of these
fluxes are have a local maximum or minimum. In contrast, the flux iso-lines cannot intersect each
other: this would mean that at the same point in the space $\{E, L\}$ the proton fluxes differ very
significantly (by an order of magnitude for the flux step selected here). Such uncertainty does not
have a physical sense and means a complete discrepancy and contradiction to each other of two
close series of data obtained in different experiments. In this case, a special analysis is needed to
identify all the errors and differences in the conditions for different measurements, in order to
reconcile them with each other.
The largest errors of our method are connected with drawing iso-lines of fluxes along
heterogeneous sets of experimental points (the errors of these points themselves do not exceed of
the size of the icons on Figs. 1–6). This uncertainty of our work is open for constructive criticism,
and these figures themselves are open for possible corrections and additions.
The synthesis of the experimental data on the fluxes of ERB ions in other representations (in
other spaces of variables) leads to more significant methodological errors and uncertainties. For
example, one can present ion fluxes obtained in various experiments for different energy channels
depending on $L$ (radial profiles of fluxes). However, the sets of these channels are different in
different experiments. To compare the radial profiles of ion fluxes for different experiments, it is
necessary to bring these fluxes to the same set of energy values. This can made by energy spectra,
but due to discreteness of these spectra, the procedures of their approximation, interpolation, and
extrapolation are inevitable. But this work can be done in different ways. With that, methodical
errors and uncertainties in the final picture acquire a hidden form. They can be tracked only if
consistently, step by step, repeating all these procedures for the experimental data.

## 2.1  Spatial-energy structure of the proton fluxes

There is a great quantity of the experimental data on the ERB protons. The most important of them
are presented in the space $\{E, L\}$ on Figs. 1 and 2. These figures are needed here for comparison
with similar distributions of ions with $Z \geq 2$ on the Figs. 3–6.
Figure 1 represent a results of the satellites Relay-1 (Freden et al., 1965); Ohzora or EXIS C:
Exospheric Satellite C, Akebono or EXOS-D: Exospheric Satellite D and ETS-VI: Engineering Test



Satellite (Goka et al., 1999). These results were obtained near minima between 19[th] and 20[th]
(1963), 21[th] and 22[th] (1984–1985), and 22[th] and 23[th] (1994–1996) of the solar activity cycles.
Figure 2 represent a results of the satellites 1968-81A (Stevens et al., 1970), Injun-5 or
Explorer-40 (Krimigis, 1970; Venkatesan and Krimigis, 1971; Pizzella and Randall, 1971), 1969-
025C or OV1-19: Orbiting Vehicle 1-19 (Croley et al., 1976), Azur or GRS A: German Research
Satellite A (Hovestadt et al., 1972; Westphalen and Spjeldvik, 1982), Molniya-1 (Panasyuk and
Sosnovets, 1973), GEOS-2: Geodetic Earth Orbiting Satellite 2 (Wilken et al., 1986), CRRES: The
Combined Release and Radiation Effects Satellite (Albert et al., 1998; Vacaresse et al., 1999), GEO-
3: Geostationary Orbit 3 (Selesnick et al., 2010) and Van Allen Probes (Selesnick et al., 2014,
2018). These results were obtained near maxima of solar activity in 20[th] (1968–1971), 22[th] (1990–
1991), 23[th] (2000), and 24[th] (2012–2017) solar cycles.
The data of the satellites Explorer-45 (Fritz and Spjeldvik, 1979, 1981) and ISEE-1:
International Sun-Earth Explorer 1 or Explorer-56 (Williams, 1981; Williams and Frank, 1984) are
given in Figs. 1 and 2 at $L > 2.5$ where solar-cyclic variations of the ERB proton fluxes are
practically do not observed (see, e.g., Vacaresse et al., 1999).
Other experimental data on ERB protons could be added to these results, but they do not change
the general picture shown in Figs. 1 and 2.
From a comparison of Figs. 1 and 2 one can see that at $L < 2.5$ (especially at $L < 1.4$) the proton
fluxes in the minima of solar activity (Fig. 1) are higher than in the maxima of solar activity (Fig.
2). In addition, in the minima of solar activity the inner edge of the proton belt is less steep and
achieve smaller $L$ shells (for $E > 1$ MeV). The distribution functions of protons $f(\mu, K, L)$ in the
phase space constructed from Figs. 1 and 2 confirm these conclusions.
In Figs. 1 and 2, the iso-lines of proton fluxes are almost parallel to each other on $L > 3$ at
sufficiently high energies. Since these iso-lines are separated from each other by approximately
equal intervals on a logarithmic scale of the energy, this region in the space $\{E. L\}$ corresponds to
power-law spectra of the ERB protons. In these figures, this region is located between the green
and red lines.
The red line is corresponded to the lower boundary ($E_b$) of the power-law tail of the proton
spectra. For this line, $E_b \sim 36 \times L^{-3}$ MeV. Some changes in the slope of these iso-lines at $L > 6$ can
be connected with an essential distinction of the magnetic field in this region from the dipole
configuration (the $L$ shells were calculated for a dipole field).
For the dipole magnetic field region, the points on the red line correspond to particles with a
specific value of the 1[st] adiabatic invariant of motion ($\mu_b$). For Figs. 1 and 2, the average value $\mu_b$
is $\sim 1.16$ keV nT$^{-1}$. Segments of an iso-lines that are parallel to the red line also correspond to a
certain values of the invariant $\mu$. In this region of the space $\{E. L\}$ the ionization and other losses
of the ERB protons during radial drift can be neglected, and the fluxes changes with changing of $L$
are practically reduced to adiabatic transformations the fluxes in a magnetic field.
It is results from these figures that at $L = 3–6$ the value $\gamma = 4.8 \pm 0.5$. At $L > 6$ the distances
between these iso-lines are increased with $L$, and the value $\gamma$ is decreased from $\sim 4.7–5.0$ at $L = 6$
to $\sim 4.1–4.5$ at $L = 8$. This is due to the deviation of the magnetic field from the dipole
configuration as well as to the increasing variability of this field with increasing $L$.
According to the data of satellites considered in (Kovtyukh, 2001), invariant parameters $\mu_b$ and $\gamma$
were found only at $L > 3$. In this work is considered the wider range of $L$ and $E$, and for protons with
$E > 10$ MeV these parameters can be traced to $L \sim 2$. At $L = 2$, $\gamma = 4.4 \pm 0.6$ (Fig. 1) and $\gamma = 4.7 \pm 1.3$
(Fig. 2). This is due to that the energy range here is significantly extended toward higher energies (up
to 200 MeV), but with increasing the energy of the ERB protons the ionization losses are decreased
rapidly (see, e.g., Schulz and Lanzerotti, 1974; Kovtyukh, 2016a).

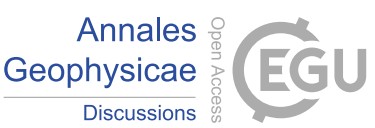

## 2.2 Spatial-energy structure of the helium ion fluxes

In Figs. 3 and 4 are presented helium ions fluxes averaged for quiet periods (Kp < 2).

Figure 3 represent the data of the satellites Molnija-2 (Panasyuk et al., 1977), Prognoz-5 (Lutsenko and Nikolaeva, 1978), ISEE-1: The International Sun-Earth Explorer 1 (Hovestadt et al., 1981); Akebono or EXOS-D: Exospheric Satellite D and ETS-VI: Engineering Test Satellite (Goka et al., 1999). These results were obtained near minima between 20[th] and 21[th] (1975–1977) ), between 21[th] and 22[th] (1984–1985), and between 22[th] and 23[th] (1994–1996) of the solar activity cycles.

Figure 4 represent the data of the satellites OV1-19: Orbiting Vehicle 1-19 (Blake et al., 1973; Fennell and Blake, 1976), Explorer-45 (Fritz and Spjeldvik, 1978, 1979; Spjeldvik and Fritz, 1981), SCATHA: Spacecraft Charging At High Altitudes (Blake and Fennell, 1981; Chenette et al., 1984). These results were obtained near maxima of solar activity in 20[th] (1968–1971) and 21[th] (1979) solar cycles.

From a comparison of Figs. 1–2 with Figs. 3–4 one can see that at $L > 2$ for helium ions the solar-cyclic (11-year) variations are greater than for protons. For example, at $L \sim 2–3$ from maximum to minimum of solar activity fluxes of protons with $E > 1$ MeV practically do not changed, and the fluxes of helium ions with $E > 1$ MeV/n are increased by one order of magnitude.

Figures 3 and 4 have the same patterns are observed as for protons, but the distribution of helium ion fluxes is slightly shifted away from the Earth (with respect to protons). Unlike protons, there are significant "white spots" for helium ions in this picture.

The red line on these figures is corresponded to the lower boundary of the power-law tail of the helium ions spectra. For this line, $E_b/M_i \sim 43.4 \times L^{-3}$ MeV/n (Fig. 3) and $E_b/M_i \sim 21.7 \times L^{-3}$ MeV/n (Fig. 4). If one take into account that for helium ions with $E > 0.2$ MeV/n at $L < 6$ the average charge $Q_i$ is +2 (see, e.g, Spjeldvik, 1979), then for the considered boundary we get: $\mu_b \sim 1.4 \times Q_i$ keV/n×nT$^{-1}$ at the maximum of solar activity and $\mu_b \sim 1.4 \times M_i$ keV/n×nT$^{-1}$ at the minimum of solar activity (for the dipole magnetic field region). The iso-lines of helium ion fluxes in Figs. 3 and 4, which pass above the red line at $L > 2.5$ are corresponded average value of $\gamma \sim 5.5$.

For helium ions, as for protons, the values of the parameters of the power-law tail of their spectra are approximately in the middle of the ranges of these parameters, which were obtained by other methods (Kovtyukh, 2001).

At the same time, one can see that the isolines of the fluxes of helium ions in the region above the red line (in the region of power-law spectra) have at a substantial slope to the red line. At $L > 3$ the fluxes of helium ions with given energy are increased with decreasing $L$ more slowly than follows from adiabatic transformations of the fluxes. In accordance to well-known calculations (see, e.g., Schulz and Lanzerotti, 1974), this is means that the ionization losses of the ERB helium ions significantly exceed these losses for protons.

## 2.3 Spatial-energy structure of the CNO group ions fluxes

In Figs. 5 and 6 are presented CNO group ions fluxes averaged for quiet periods (Kp < 2).

Figure 5 represent the data of the satellites ATS-6: Applications Technology Satellite 6 (Spjeldvik and Fritz, 1978; Fritz and Spjeldvik, 1981) and ISEE-1: The International Sun-Earth Explorer 1 (Hovestadt et al., 1978). These results were obtained near the minimum between 20[th] and 21[th] of the solar activity cycles (1974–1975, 1977).

Figure 6 represent the data of the satellite Explorer-45 (Spjeldvik and Fritz, 1978; Fritz and Spjeldvik, 1981). These results were obtained near the maximum of solar activity in 20[th] solar cycle (1971–1972).

On Figs. 5–6 the spatial-energy patterns of the fluxes for the CNO group ions is even more shifted away from the Earth and its configuration differ significantly from the Figs. 1–4.

From a comparison of Figs. 1–2 with Figs. 5–6 one can see that for ions of CNO group the solar-cyclic (11-year) variations are greater than for protons. For example, at $L \sim 3–5$ from





maximum to minimum of solar activity fluxes of protons with $E > 1$ MeV practically do not
changed, but the fluxes of the CNO group ions are increase by one order of magnitude and more.
From a comparison of Figs. 3–4 with Figs. 5–6 it is seen also that the fluxes of CNO group ions
varies by several times greater than the fluxes of helium ions.
This is mean that for ions of the CNO group the ionization losses at $L = 3$–$5$ are much larger
than for ions with $Z \leq 2$ and these losses have a significant effect even on the power-law segment
of the spectra of the CNO ions (in the part which is seen on Figs. 5–6). Therefore, the lower
boundary of the power-law tail of these ions spectra is not monitored on the data given in Figs. 5
and 6. The red line on these figures is rather arbitrary: it corresponds to adiabatic laws that are not
performed here, but this line let us to trace these deviations. As can be seen from fig. 5–6,
ionization losses for ions of the CNO group are especially large at the maximum of solar activity
(Fig. 6): in these times the slope of iso-lines on $L > 3$ is significantly less than the slope of the red
line.
At the same time, at $L > 4$ in Fig. 5 and at $L > 3$ in Fig. 6 the iso-lines of fluxes pass almost
parallel to each other and at approximately equal distances from each other; the average value of $\gamma$
corresponding to them is $\sim 6$. Thus, for sufficiently large values of $E$ and $L$, the CNO group ions
spectra in the ERB have a power-law form, but these spectra are softer in comparison with the
spectra of protons.
The red line corresponds here to the dependences $E_b/M_i \approx 43.4 \times L^{-3}$ MeV/n (on Fig. 5) and $E_b/M_i$
$\sim 12.4 \times L^{-3}$ MeV/n (on Fig. 6), which are taken from (Kovtyukh, 2001) where this boundary was
more clearly defined also for the ions of the CNO group. If one take into account that for the CNO
group ions with $E > 0.1$ MeV/n at $L \sim 3$–$5$ the average charge $Q_i$ is $+4$ (see, e.g., Spjeldvik and
Fritz, 1978), then for this boundary one can get: $\mu_b \sim 1.4 \times Q_i$ keV/n×nT$^{-1}$ at the maximum of solar
activity and $\mu_b \sim 1.4 \times M_i$ keV/n×nT$^{-1}$ at the minima of solar activity (for the dipole magnetic field
region).

## 286    3   Discussion

Let us consider the conclusions following from the results obtained here for solar-cyclic variations
in the fluxes of ERB ions. Solar-cyclic (11-year) variations of proton fluxes with $E > 1$ MeV in the
inner region of the ERB considered in many works (see, e.g., Pizzella et al., 1962; Hess, 1962;
Blanchard and Hess, 1964; Filz, 1967; Nakano and Heckman, 1968; Vernov, 1969; Dragt, 1971;
Huston et al., 1996; Vacaresse et al., 1999; Kuznetsov et al., 2010; Qin et al., 2014). These
variations achieve one order of magnitude at $L = 1.14$ and reduce rapidly with increasing $L$ (see,
e.g., Vacaresse et al., 1999).
In these works, such variations of the proton fluxes of the inner belt are connected to the solar-
cyclic variations of the energy loss rates of protons in this region. However, solar-cyclic variations
of fluxes of ions with $Z \geq 2$ were not considered in these works.
In quiet periods, only the mechanism of ionization losses is significant for the ERB protons
trapped on small $L$ shells (see, e.g., Schulz and Lanzerotti, 1974). Energy loss rates and lifetimes of
the ERB protons are determined in this mechanism by the density of atmospheric atoms and
ionospheric plasma ($N$) in a geomagnetic trap. This density is depend on the intensity of the
ultraviolet radiation of the Sun.
With decreasing solar activity (with a transition from maximum to minimum of the solar cycle),
the densities of atmospheric atoms and ionospheric plasma in a geomagnetic trap are decreased. If
the proton supply rates to the inner belt under the action of the CRAND mechanism remain
unchanged or the effect of these changes is weaker than the effect connected with changes of loss
rates of the protons, the stationary proton fluxes will increased with the solar activity decreasing.
The lifetimes of protons increase with $L$, and it lead to decreasing in the amplitude of the solar-
cyclic variations of a proton fluxes. The proton lifetime on a given $L$ shell depends on their energy





and is less than 11 years at $L < L^*(E)$. For example, for protons with $E \sim 10$ MeV the value $L^*$ is ~
2.5 (see, e.g., Kovtyukh, 2016a). Figs. 1 and 2 show that for protons the solar-cyclic variations of
fluxes are small and localized at $L < 2.5$ (mainly at $L < 1.4$).
In contrast to protons, Figs. 3–6 show significant solar-cyclic variations of fluxes of helium ions
and CNO group ions at $L \sim 2$–5. These variations one can explain by the same mechanism, which
suggested for proton fluxes at $L < 2.5$.
For ions with $Z \geq 2$ in the ERB ionization losses are more significant than for protons. With this
fact one can connect the absence of ions with $Z \geq 2$ at $L < 2$ (or very low values of these fluxes)
during quiet geomagnetic conditions. More short lifetimes of these ions compare to protons are
manifested also in the slope of the experimental curves in Fig. 4 and 6 (this was noted in sections
2.2 and 2.3, respectively). Consequently, for ions with $Z \geq 2$ the regions in which the solar-cyclic
variations can manifested should be located on higher $L$ shells (at the same energies as for
protons).
The lifetimes of ions of the energies considered here are $\tau \propto M_i^{-1/2} Q_i^{-2} N^{-1} E^{3/2}$ (Schulz and
Lanzerotti, 1974). In a first approximation, for $N \propto L^{-4}$, we obtain the value $L_i^* \sim (M_i^{1/2} Q_i^2)^{1/4} L^*$,
where $L^*$ corresponds to protons of the same energy as other ions. For helium ions ($M_i = 4$, $Q_i = 2$)
with $E \sim 10$ MeV, we obtain $L_i^* \sim 4.2$. For ions of CNO group ($M_i = 14$, $Q_i = 4$) with $E \sim 10$ MeV we
obtain $L_i^* \sim 6.9$. These is very rough estimates, but they correspond to results presented in Figs. 3–6.
These estimates are based on the following assumption: during variations in solar activity, the rates
of ion supply on $L < L_i^*$ remain unchanged (or these changes are weaker than the effect of changes of
the rate of ion losses). This assumption is real for protons with $E > 10$–20 MeV at $L < 2.2$: the fluxes
of these protons forming mainly under the action of the CRAND mechanism. However, at $L > 2.2$ the
stationary ion fluxes of the ERB forming mainly under the action of radial diffusion (see, e.g., Schulz
and Lanzerotti, 1974; Kovtyukh, 2016b, 2018). Therefore, the solar-cyclic variations of fluxes for ions
with $Z \geq 2$ one can explain only under the assumption that the effect connected with an increasing in
the ionization losses of such ions significantly exceeds the effect connected with the possible enhance
of radial diffusion of ions on the growth phase of solar activity.
In the experimental results presented here for the ERB ions, the region of the power-law tail of
the ion spectra is distinguished. For many experiments, especially for heavy ions, the values of the
parameter of a power-law tail spectra are determined much more accurately by the dependences
$J(L)$ of the ion fluxes (in the logarithmic scale) for different pairs of energy channels (see
Kovtyukh, 2001). For example, the range $L$ in which these dependences for two energy channels
are parallel to each other is corresponded to the power-law tail of the spectra. On smaller $L$ these
fluxes begin to converge and radial dependences of the fluxes intersect with each other which is
corresponded to the maximum in the spectra. Consider here the physical mechanisms leading to
the formation of power-law distributions of ions of the ERB.
The main source of ions in the outer regions of the ERB is the solar wind, and the high-energy
part of these spectra have an exponential shape usually (see, e.g., Ipavich et al., 1981a, 1981b).
Immediately before being captured into the magnetosphere, these ions pass through a highly
turbulized regions, but the high-energy part of their spectra usually retains an exponential shape.
Therefore, the question arises: what physical mechanism converts the form of ion spectra from
exponential to power-law?
Evidently, the power-law tail of the ERB ions spectra must be generate-in the outer regions of
the magnetosphere. The most likely region for this is the plasma sheet (PS) of the magnetospheric
tail, which is adjacent to the geomagnetic trap. High-energy part of the ion spectra in the PS, at $R \sim$
20–40 $R_E$, have power-law shape and the exponents of these spectra is close to the corresponding
parameters of the spectra of ions in the ERB. On the data of the satellites IMP-7 and IMP-8:
Interplanetary Monitoring Platform 7 and 8 (Sarris et al., 1981; Lui et al., 1981) and also satellite
ISEE-1 (Christon et al., 1991), the shape of the ion spectra of the PS usually do not changed during





substorms; they produce only parallel shifts of the spectra along logarithmic axes $E$ and $J$. These
results point out that the time scales of formation processes of these ion spectra in the PS are far
exceeds the times of substorms.
Parameters of the power-law tail of the ion spectra of the outer belt ($\gamma$ and $\mu_b$) reflect,
apparently, the most fundamental features of the mechanisms of acceleration of ions in the tail of
the magnetosphere. One can try to connect the values of these parameters with the most general
representations about the mechanisms and character of ion acceleration in the PS of the
magnetospheric tail.
Most likely, this part of the ion energy spectra formed in the PS by stochastic mechanisms of
the ion acceleration. This hypothesis supported by many experimental results. Statistical character
of these mechanisms reveal itself, in particular, in the fact that the ratios of fluxes (and partial
densities) of ions with different $Z$ can be differ greatly at low and high energies. During wander in
the phase space, ions gradually forget their origin and, therefore, the high-energy tails of the ion
spectra do not contain unambiguous information on the partial densities of different components of
ions in the source (see, e.g., Kovtyukh, 2001).
The high-energy part of the ion spectra of the PS can be generated by the mechanisms of
acceleration of particles on magnetic irregularities moving relative to each other (Fermi
mechanism). The fractal structures of the PS reveal itself on scales from ~ 0.4 to ~ 8 thousands
kilometers, for example, in the data of the satellite Geotail (Milovanov et al., 1996). If mass of the
ions are small compare to masses of the magnetic irregularities in the PS, the average values of the
index $\gamma$ of the power-law tail of the spectra should not depend on mass and charge of these ions.
Under equilibrium conditions, this parameter is determined by the average part of energetic ions
in the total energy density of particles and magnetic irregularities ($\bar{\beta}$). From the theory which was
developed by Ginzburg and Syrovatskii (1964), it is follows: $\gamma - 1 \approx (1 - \bar{\beta})^{-1}$. With increasing $\bar{\beta}$
in the interval $0 < \bar{\beta} < 1$, the value $\gamma$ is increase monotonically and $\gamma \to \infty$ for $\bar{\beta} \to 1$. For real
average values $\bar{\beta}$ in the central PS, we get $\gamma \sim 3.5$–7.0 ($\gamma \sim 4.3$ at $\bar{\beta} \sim 0.7$).
Spectra with power-law tail and quasi-exponential segment at lower energies can be generated
when the value $\Delta B / \bar{B}$ for magnetic irregularities is proportional to their size $\delta r$ and the spectral
density of irregularities is decrease rapidly with increasing $\delta r$ for $\delta r < r_s$, but for $\delta r > r_s$ it remains
almost unchanged. Apparently, the spectra of magnetic irregularities in PS with thickness $r_s$ have
just such form. Then the lower boundary $\mu_b$ of the power-law tail is corresponded to the condition
$r_s/\rho_i \sim 10$ ($\rho_i$ is the gyroradius of ions), i.e. $\mu_b \sim 0.02(Q_i^2/M_i)B_s r_s^2$ keV nT$^{-1}$, where $B_s$ is the average
magnetic field induction in the PS (in nT) and $r_s$ is normalized to the Earth's radius. Believing that
$B_s \sim 30$ nT and $r_s \sim 1.3$ R$_E$ it can be obtained: $\mu_b \sim 1.0$ ($Q_i^2/M_i$) keV nT$^{-1}$.
The energy spectra of ions in the radiation belts of such planets as Jupiter and Saturn have the
form analogous to the form of ion spectra in the ERB (see, e.g., Krimigis et al., 1981; Cheng et al.,
1985). As that in the ERB, these spectra have a long power-law tail, which is formed, apparently,
by mechanisms of stochastic acceleration of ions as a result of interactions of these ions with the
current layer of the magnetospheric tail.

## 5   Conclusions

There are analyzed the experimental results for the stationary fluxes of the main ion components of
the ERB (protons, helium ions and ions of the CNO group) in the near equatorially plane. It is
found that in the outer belt these fluxes line up in the certain regular patterns in the space $\{E, L\}$.
The degree of such similarity is increase with increasing $E$ and $L$. The similarity of the spatial-
energy distributions for various ionic components of the ERB is based on the main sources and on
the universality mechanisms of transfer, acceleration and losses of ERB ions in the outer belt
(radial diffusion while conserving $\mu$ and $K$ of ions, betatron acceleration and ionization losses).



Solar-cyclic (11-year) variations of the spatial-energy distributions of the ERB ion fluxes are
investigated. It is find that the ERB ions fluxes are weaken with increasing the solar activity and
this effect increases with increasing an atomic number $Z$ of the ions. Such a dependence of the
amplitude of flux changes on $Z$ is typical also for faster variations in the fluxes of the ERB ions,
during geomagnetic storms and other disturbances of the Earth's magnetosphere, what is
underlined in the review Kovtyukh (2018).
The figures presented here make it possible to determine in which regions of the space $\{E, L\}$
near the equatorial plane the ionization losses of ions during their radial diffusion can be neglect
and where this cannot be done. These results indicate also that with variations in the level of solar
activity the coefficient $D_{LL}$ of the radial diffusion of the ERB ions change much less than the
ionization losses rates of ions with $Z \geq 2$.
In addition, the figures given here reveal the localization of "white spots", especially extensive for
ions with $Z \geq 2$ and $E > 1$ MeV/n at $L < 3$. The larger $Z$ and energy of ions and the smaller $L$ the
greater the uncertainty in the values of the ERB fluxes. These gaps must be filled by the results of the
future experiments on the satellites. Now the extensive gaps in the experimental data for fluxes of ions
with $Z \geq 2$ do not allow create the sufficiently complete and reliable empirical models of the ERB for
these ions.

## Acknowledgements.

**Financial support**. This work was supported by Russian Foundation for Basic Research RFFI
grant No. 17-29-01022.

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



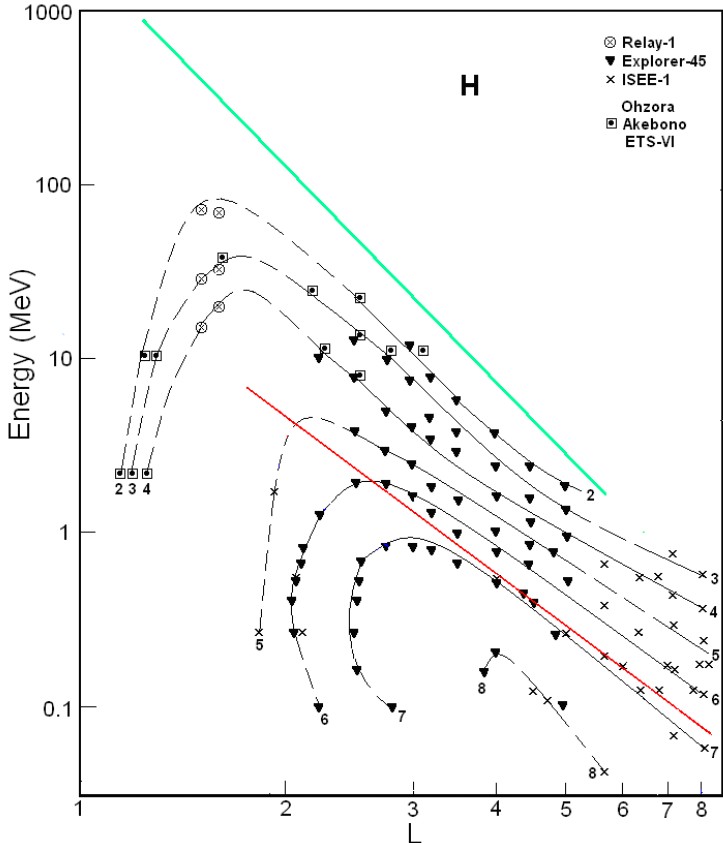

**Figure 1.** Proton fluxes in the ERB near minima of the solar activity. A numbers on the curves are equal the values of
the decimal logarithms of $J$ where $J$ is given in $(cm^2 \ s \ ster \ MeV)^{-1}$; it is differential fluxes of protons with $\alpha_0 \approx 90^o$
(near the plane of the geomagnetic equator). Data of satellites are presented by different symbols. The red line
corresponded to the lower boundary of the power-law tail of the proton spectra; green line corresponded to the
maximum energy of protons trapped in the ERB (Ilyin et al., 1984).

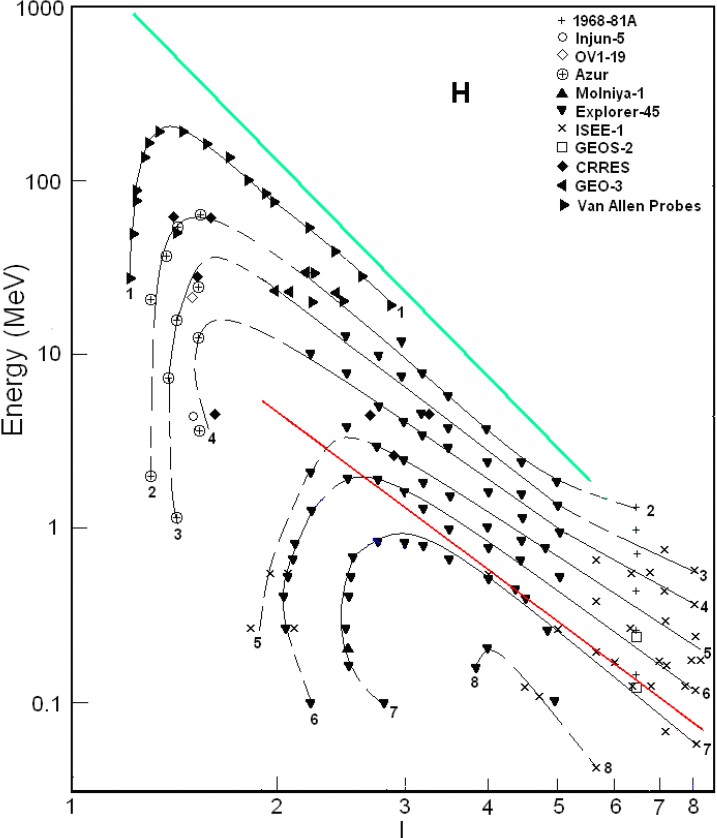

**Figure 2.** Proton fluxes in the ERB near maxima of the solar activity. A numbers on the curves are equal the values of
the decimal logarithms of $J$ where $J$ is given in (cm$^2$ s ster MeV)$^{-1}$; it is differential fluxes of protons with $\alpha_0 \approx 90^{\circ}$
(near the plane of the geomagnetic equator). Data of satellites are presented by different symbols. The red line
corresponded to the lower boundary of the power-law tail of the proton spectra; green line corresponded to the
maximum energy of protons trapped in the ERB (Ilyin et al., 1984).

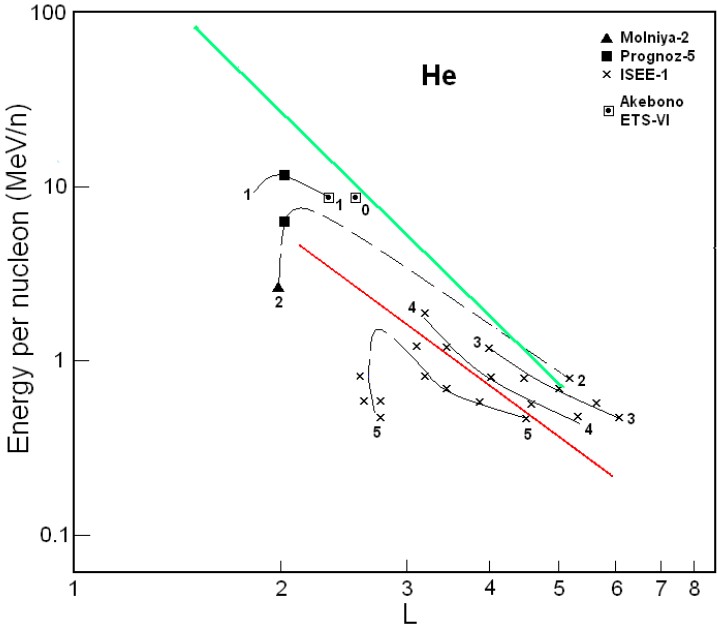

**Figure 3.** Helium ion fluxes in the ERB near minima of the solar activity. A numbers on the curves are equal the values of the decimal logarithms of $J$ where $J$ is given in $(cm^2 \text{ s ster MeV/n})^{-1}$; it is differential flux of protons with $\alpha_0 \approx 90^o$ (near the plane of the geomagnetic equator). Data of satellites are presented by different symbols. The red line corresponded to the lower boundary of the power-law tail of the helium spectra; green line corresponded to the maximum energy of these ions trapped in the ERB (Ilyin et al., 1984).

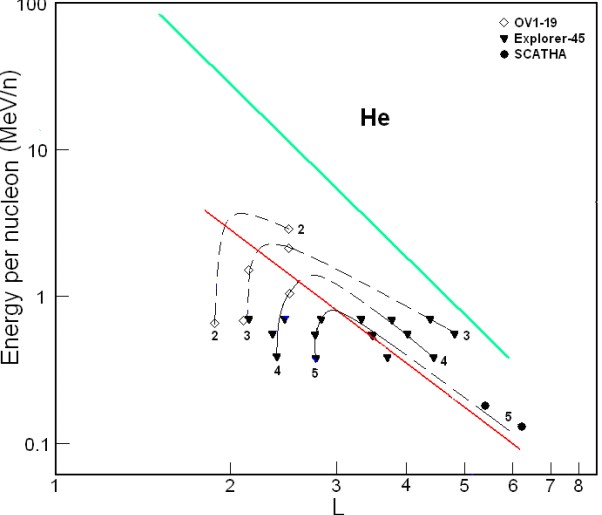

**Figure 4.** Helium ion fluxes in the ERB near maxima of the solar activity. A numbers on the curves are equal the value of the decimal logarithms of $J$ where $J$ is given in $(cm^2 \text{ s ster MeV/n})^{-1}$; it is differential flux of protons with $\alpha_0 \approx 90^o$ (near the plane of the geomagnetic equator). Data of satellites are presented by different symbols. The red line corresponded to the lower boundary of the power-law tail of the helium spectra; green line corresponded to the maximum energy of these ions trapped in the ERB (Ilyin et al., 1984).



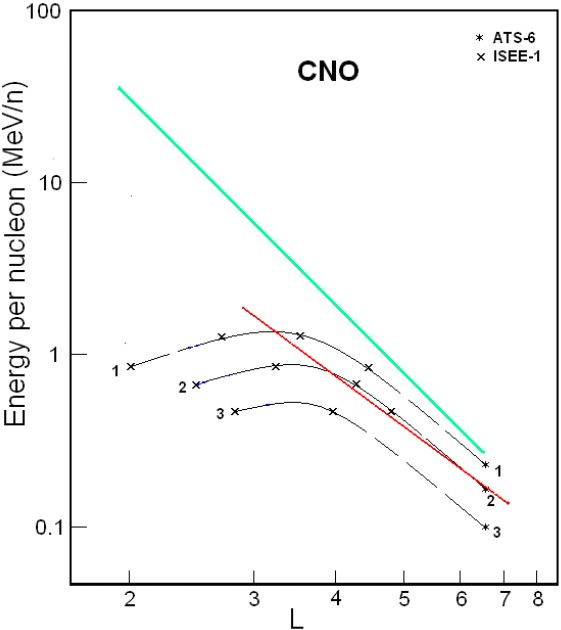

**Figure 5.** CNO ion fluxes in the ERB near minima of the solar activity. A numbers on the curves are equal the values
of the decimal logarithms of $J$ where $J$ is given in $(cm^2 \, s \, ster \, MeV/n)^{-1}$; it is differential flux of protons with $\alpha_0 \approx 90^o$
(near the plane of the geomagnetic equator). Data of satellites are presented by different symbols. The red line
corresponded to the lower boundary of the power-law tail of the CNO ion spectra; green line corresponded to the
maximum energy of these ions trapped in the ERB (Ilyin et al., 1984).

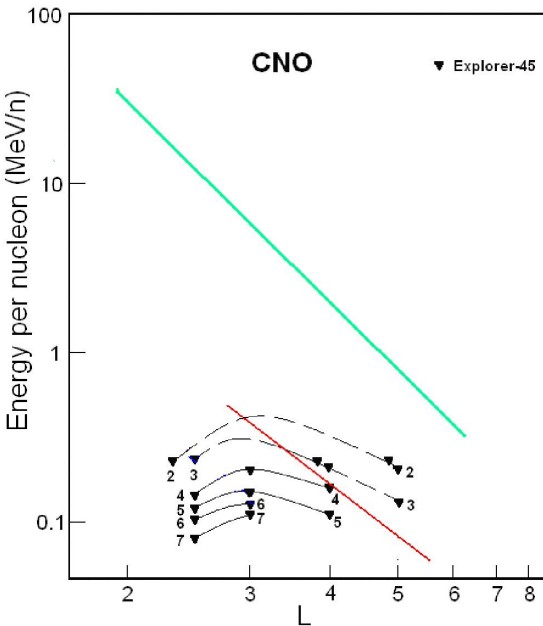

**Figure 6.** CNO ion fluxes in the ERB near the maximum of the solar activity. A numbers on the curves are equal the
values of the decimal logarithms of $J$ where $J$ is given in $(cm^2 \, s \, ster \, MeV/n)^{-1}$; it is differential flux of protons with $\alpha_0$
$\approx 90^o$ (near the plane of the geomagnetic equator). Data of satellites are presented by different symbols. The red line
corresponded to the lower boundary of the power-law tail of the CNO ion spectra; green line corresponded to the
maximum energy of these ions trapped in the ERB (Ilyin et al., 1984).