# Peer review of "Earth's radiation belts ions: Patterns of the spatial-energy structure"

_Annales Geophysicae, 2019_

## Referee Comment (RC1) · Anonymous Referee #1 · 22 Nov 2019

The manuscript addresses the scientific topic of understanding the spatial distributions of protons, Helium nuclei and CNO-group ions in the Earth Radiation Belts together with their variations during various solar minima and maxima. Even if it does not contain new data (it analyzes a wide range of older data from satellites), it offers a new interpretation of the mechanisms behind the different distributions of particle populations inside the magnetosphere. The exposition is fairly linear and the theoretical approach is sufficiently explained by the author, even if the description of the 3 sections seems rather repetitive, but it is something due to the very nature of the manuscript. The abstract reflects the content of the manuscript very well. Overall, it appears a little too long and with quite a few grammatical errors that make reading pretty hard and not fluent. Nevertheless, the topic is interesting, figures are simple but explanatory and the conclusions are satisfactory; thus, in my opinion, could be published after some fairly substantial grammar revision.

I have a few more comments, then I will provide a table with some suggestions on how to correct some grammatical mistakes.

1. lines 96-97: what does "averaged" means here? And what does it mean that "All values of differential fluxes reduced to one dimension"? I think that this sentence should be rewritten more clearly

2. lines 144-147: this whole sentence, in my opinion, is not suitable for a scientific paper. The method used is the result, I believe, of your careful reasoning. Therefore the scientific community will evaluate and judge it on its own. You do not have to show any doubts about your choice. You can just address the fact that the uncertainties are linked to the errors of the experimental points shown in the various Figures.

3. lines 148-157: in my opinion, describing in too much detail a procedure or a method that has not been used in the paper, does not help the linearity of the text and distracts the reader. This whole paragraph could be restricted to just two sentences. For example "[...] Representing plots in a different space of variables would lead only to more significant methodological errors and uncertainties, because of the natural differences in the instrumentation of the experiments taken into account; thus, a series of approximations or interpolation/extrapolation techniques would become inevitable.".

4. lines 179-180: I would remove this sentence because it does not add much to the description of the Figures.

5. line 184: the function $f(\mu, K, L)$ is not previously cited nor defined. I guess it is a reference to line 30 and to the curves in the {E,L} plane, but maybe this implicit formula could be at least introduced earlier in the text.

6. line 201: What is $\gamma$? I can infer it is the slope of the flux (spectral index) but maybe in this occasion, it could be useful to, at least, describe (with a couple of words) what it means explicitly in these plots.

| l p12cm | | |
|---|---|---|
| Line | Comment | |
| 1 | I think that after ":" the P in "Pattern" should be lowercase | |
| 6 | In my opinion the word "ions" appears two times in too little space. I would write "[...] protons, helium and ions of [...] " | |
| 8-9 | "[...] considered here using data from satellites in the period 1961–2017" | |
| 9-10 | "It has been found that the results of these measurements line up [...] following some regular patterns" | |
| 10-11 | see line 6 | |
| 11-14 | This sentence is a little bit convoluted and it may result difficult for the reader to understand. I would suggest: "[...] It has been observed that in the inner regions of the ERB, ion fluxes decrease with increasing solar activity and that the solar-cyclic variations of fluxes of $Z \geq 2$ ions are much greater than for protons; moreover, it seems that they increase with increasing atomic number Z." | |
| 14 | "Finally, the possible physical [...] " | |

21-22  "The ERB consist mainly of [...] "

23-24  The use of $Z \geq 2$ here is redundant, in my opinion. Helium and Oxygen are already ions with $Z \geq 2$ so I think that you can modify the sentence as "In ERB there are also helium nuclei and other $Z > 2$ ions (like Oxygen etc) [...] "

"[...] disturbances, ion fluxes, and their distributions are changed"

I think that "[...] between a local vector [...] " should be "[...] between the local vector [...] " because in a well-determined region you can have only one specific vector of the magnetic field. Moreover, I would put a "the" also in front of "vector of a particle velocity"

"which are injected [...] "

"[...] drifted conserving [...] "

"This layer is called the drift shell"

32-33  I will suggest adjusting the following sentence as follows, to maintain the text more easily readable: "Therefore, experimental data on the ERB are often represented in coordinates {L, B}, where L is the drift shell parameter and B is the local induction of the magnetic field [...] "

"center of the dipole itself (in Earth's radii $R_E$)"

In my opinion, the use of the passive form here should be avoided: "fluxes [...] decrease [...] "

"[...] to higher latitudes [...] "

I would add a "respectively" at the end of the sentence, to better distinguish between B and $B_0$

42-43  "Outer and inner regions of the ERB are maintained in dynamic equilibrium with the environment by different mechanisms "

44-46  I would suggest to slightly modify this sentence as follows: "The outer belt [...] is formed mainly by the mechanisms of radial diffusion of such ions towards the Earth under the action of fluctuations of both electric and magnetic fields resonating with their drift periods"

46-48  I would avoid the repetition of the word "ions" here. Moreover, I would re-write these parts: "This transport is accompanied [...] " and "as a result of their interactions [...] "

49-53  I think that there is a temporal mismatch between "[...] protons with $E > 10$ MeV is formed [...] " and "The inner belt of ions with $Z > 4$ was formed [...] "

54-55  "[...] the mechanism of ion capture from Solar Cosmic Rays takes place [...] "

"[...] ERB, together with the sources of injection and losses of ions, are known"

58-61 The sentence here is very important, in my opinion, because it introduces the problem that this paper tries to solve, so it should be written more clearly. "However, for comprehensive verification of the physical models and to identify the mathematical models and their parameters, the formulation of complete and reliable empirical models of the ERB for each of the ion components, is necessary; it is also necessary for ensuring the safety of space flights [...] "

62-67 "These models can be created only using experimental data, obtained over many years and decades; such models [...] were already created for protons (AP8/AP9) and they are widely used in space research. On the contrary, measurements of $Z \geq 2$ ion fluxes suffer from technical problems due to small statistics and high background of protons and electrons. For this reasons, empirical and semi-empirical models for $Z \geq 2$ particles, are applicable [...] "

68-72 I think that the first part of this sentence is just a repetition of what already said in lines 62-67; therefore I would leave only the part "One of the main problems of this work is to consider the possibility to create a sufficiently complete and reliable [...] "

73-76 "In the following sections, the spatial-energy structure [...] (Sect. 2) together with the possible physical [...] are considered [...] , and the main [...] are given (Sect. 4) "

78-79 The sentence here is a bit confused. I would suggest to change it as "There can be ions trapped in drift shells only with energies less than some maximum values, determined by [...] "

gyroradius

"[...] The green line in Figs 1-6 represents this very boundary [...] "

87-88 I would remove the second "satellites " here and just leave the phrase as: "[...] with the differences in their trajectories [...] "

"[...] sets of energy channels [...] "

I would substitute "Solar" with "Sun"

"[...] of the Earth during various periods of data-taking [...] "

91-92 "[...] influence the fluxes of [...] with respect to proton fluxes [...] "

94-95 "[...] In this section, experimental data of various [...] $90°$ have been used."

97-98 "[...] L shells, where these data were obtained, the ion fluxes are not distorted by the background [...] "

99-103 "In many important experiments, the instruments were not able to separate fluxes of ions by their charge. Moreover, for the ions of the CNO group, the separation by mass are not usually performed. For heavier species, for example for Fe ions, we have very small data-sets. Therefore, this work presents data on helium ions (without any charge separation) and CNO ions (without any mass or charge separation)."

104-108 "To solve the aforementioned problems, it is important to choose the form of representation (space of variables), in which the results of the single experiments can be compared to each other. In our case, the space {E. L} has been used; this choice is very efficient to better organize fragmentary experimental data obtained in different ranges of E and L."

112-113 "Experimental points on these figures are connected by lines [...] "

113-115 I think that a wide number of small sentences can break the fluency of the discussion. So, in this case, I would suggest to remove the last one and write: "[...] the decimal logarithms of the fluxes J, in unit of $(\text{cm}^2\text{s ster MeV/n})^{-1}$, are shown near each iso-lines"

"[...] but also very convenient [...] "

118-119 I think that both sentences inside the parenthesis () could be removed here.

120-143 This whole paragraph is crucial for the reader to better understand the meaning of Figures 1-6 and how they were built. So, the concept should be expressed in a more concise and linear way; therefore, I strongly suggest to review this paragraph. Meanwhile, I suggest a possible rewording: "The points in Figs. 1-6 have been obtained from the values of the single ion fluxes J(L) as a function of energy (the average energy for each channel of the instrument) and with an equatorial pitch angle close to $90°$. Unlike electron fluxes or ion fluxes measured during geo-active conditions, the ion fluxes considered here (i.e. during quiet periods), usually have only one maximum. As a result, for each experiment, 1 or 2 points were obtained (both shown on the outer/inner edges of the E vs L profiles). Sometimes, especially for low fluxes, only one point was obtained: in these cases, the radial profile of the ion fluxes was cutoff at small values of L due to a significant background of contaminating particles and no interpolation/extrapolation has been performed whatsoever. Each iso-line, shown in these figures, has been evaluated separately from the corresponding set of experimental points (icons); thus, in more abundantly populated sectors of the plots (i.e. for protons with E > 1 MeV at L > 2) such iso-lines are mixing in Figs. 1-2. In case of a large distance between neighboring points, the corresponding segments of the iso-lines are shown as dashed arcs. The radial profiles of the differential fluxes J(L) of particles with different energy tend to intersect with each other in those regions where the energy spectra present some local maxima or minima. On the contrary, the iso-lines cannot intersect with each other: because this would mean that, at the same point in the {E, L} space, proton fluxes differ very significantly (by an order of magnitude). Such uncertainty does not have a physical sense and a special analysis is needed to identify other possible sources of error."

159-161 "[...] There is a large number of experimental data concerning ERB protons; the most important of them are presented in Figs. 1 and 2. These figures serve as a comparison with similar distributions of $Z \geq 2$ ions (Figs. 3–6)."

"[...] Figure 1 represents results from the satellites [...] "

"[...] have been collected during minimum periods of various solar cycles, i. e. between [...] "

173-174 "[...] These results were obtained during maximum periods of solar cycles: $20^{th}$ (1968-1971), $22^{nd}$ (1990-1991), $20^{rd}$ (2000), and $24^{th}$ (2012-2017) [...] "

175-178 "The data [...] are given in both Fig 1 and 2 because solar-cyclic variations of the ERB proton fluxes are negligible [...] "

181-184 "[...] the proton fluxes during solar minima (Fig. 1) are higher than during maxima (Fig. 2). In addition, in the former the inner edge of the proton belt is less steep and it can reach smaller L shells [...] "

"[...] The red line corresponds to the [...] "

192-194 "[...] can be connected to a discrepancy between the real configuration of the magnetic field lines of the magnetic field and the dipolar configuration (used here for L shell calculation) [...] "

197-198 "[...] correspond to certain values [...] "

"[...] changes of fluxes with changing L [...] "

"[...] transformations in a magnetic field [...] "

"[...] It results from these [...] "

"distances [...] increase [...] "

"In this work, a wider range of L and E is considered [...] "

208-210 "[...] This is due to the fact that the energy range is significantly extended toward higher values (up to 200 MeV), but here the ionization losses for protons rapidly decrease [...] "

I would put "are presented " at the end of this sentence

"Figure 3 represents [...] "

"Figure 4 represents [...] "

215-224  For these sentences, I would suggest following the comments on lines 162-174

I suggest removing "are observed "

229-230  "[...] but the distribution of helium ions is slightly shifted towards higher values of L shell [...] "

I think that could be useful to better define "white spots". Do you mean that there are few experiments on ERB helium studies?

"corresponds to the lower [...] "

"[...] If one takes into account [...] "

"[...] , which pass above the red line at $L > 2.5$, correspond to an average value of [...] "

239-240  I would summarize a bit this sentence, writing only "[...] For helium ions spectra, as for protons ones, the values of the parameters of the power-law tail are in good agreement with what has been found in [...] "

It is not clear to me what you intend here... I guess "[...] deviate from the slope of the red line [...] " - is that correct?

244-245  "[...] increase with decreasing L slower than expected from adiabatic transformation. "

245-247  "This means that the ionization losses of the ERB helium ions significantly exceed these losses for protons, in agreement to well-known calculations [...] "

249-256  See comments on lines 162-174, 215 and 220

"[...] of the ion fluxes of the CNO group [...] "

See my comments on line 230

"[...] but the fluxes of the CNO group increase by one order of magnitude or more [...] "

263-264  "[...] it is seen also that the fluxes of the CNO group change several times more than the fluxes of helium ions do [...] "

"[...] This means that [...] "

269-270  "[...] to adiabatic laws that are not reported here, but this line let us estimate such deviations [...] "

"[...] If one takes into account [...] "

"[...] following the results [...] "

"[...] of the ERB have been studied in many works [...] "

291-292 "[...] These variations reach one order [...] "

"[...] This density depends on [...] "

"decrease "

"[...] the stationary proton fluxes will increase with the decreasing solar activity [...] "

307-308 "[...] and this leads to a decrease in the amplitude of the solar-cyclic variations of proton fluxes [...] "

"[...] depends on its energy [...] "

"[...] These variations can be explained by the same mechanism that has been suggested for protons at L < 2.5 [...] "

315-316 "[...] For ions with Z ≥ 2 in the ERB, ionization losses are more significant than for protons and this can be connected to the absence of ion [...] "

317-318 "[...] Such short lifetimes are manifested also [...] "

"[...] Z ≥ 2, the regions in which variations can manifest, should be [...] "

"[...] in the energy ranges considered here [...] "

"[...] of the same energy of the other ions under study [...] "

"[...] These are very rough estimations, but they are in agreement with the results presented in [...] "

329-330 "[...] for protons with E > 10-20 MeV at L < 2.2; in fact, protons form mainly under the action of the CRAND mechanism [...] "

"form mainly [...] "

333-335 "[...] the solar-cyclic variations of Z ≥ 2 ion fluxes can be motivated only under the assumption that the effect related with an increase in the ionization losses of such ions significantly exceeds the effect connected with the possible enhance of radial diffusion of ions on the rising phase of solar activity [...] "

"in logarithmic scale "

340-341 "[...] the range of L, in which these dependencies for two energy channels are parallel to each other is connected to the power-law tail of the spectra [...] "

341-343 "[...] Instead, on smaller values of L, these fluxes begin to converge and the radial dependencies of these fluxes intersect with each other, which is related to the maximum in the spectra. [...] "

343-345  I suggest to remove this sentence here, or at least merge it with the following one on lien 345: "Concerning the physical mechanisms leading to the formation of power-law distributions of ions, the main source of ions in the outer [...] "

"[...] usually spectra have an exponential shape [...] "

"[...] The most likely region for this to happen is the plasma sheet [...] "

"[...] the exponents of these spectra are close [...] "

I think that the acronym for IMP-7 and 8 is well known to people reading this paper, I would leave only the reference to Sarris et al.

"[...] the shape of the ion spectra of the PS usually do not change during [...] "

359-360  "[...] in the PS exceed the times of substorms [...] "

"[...] representations of the mechanisms [...] ". I would also suggest removing "and character"

"[...] this part of the ion energy spectra is formed in the PS by stochastic mechanisms of ion acceleration; this hypothesis is supported by many experimental results [...] "

367-372 "[...] The statistical aspect of these mechanisms reveals itself, in particular, in the fact that the ratios of fluxes (and partial densities) of ions with different Z can differ, even greatly, at low and high energies. During their passage in the phase space, ions gradually loose information about their origin and, therefore, the high-energy tails of their spectra contain ambiguous information on the partial densities of different components of ions in the source [...] "

373-378  The high-energy portion of the ion spectra of the PS can be generated by the mechanisms of acceleration of particles on magnetic irregularities moving with respect to each other (Fermi mechanism). The fractal structures of the PS are revealed on scales from $\sim 0.4$ to $\sim 8$ thousands kilometers, for example, in the data of the satellite Geotail [...] If the mass of the ions are small compared to the mass of the magnetic irregularities in the PS, the average values of the index $\gamma$ of the power-law tail should not depend on mass and charge of such nuclei "

"it follows "

"[...] $<1$, $\gamma$ increases monotonically [...] "

385-386  "[...] and their spectral density decreases rapidly with increasing [...] "

"corresponds to the condition [...] "

"[...] form analogous to that of ion spectra in the ERB [...] "

398-399  "[...] In this work, the experimental results [...] plane, have been analyzed [...] "

401-404  "[...] The degree of such similarity increases with [...] and it is linked to the nature of the main sources and on the universality [...] "

405-407  "[...] Moreover, solar-cyclic (11-year) variations of the spatial-energy distributions of the ERB ion fluxes have been investigated, It has been noted that the ERB ions fluxes are weaker with increasing solar activity and this effect increases with increasing atomic number Z. "

"[...] is typical, also, for faster [...] "

409-410  "[...] as has been underlined in [...] "

"[...] their radial diffusion can be neglected [...] "

417-418  "[...] As Z and energy become larger and L becomes smaller, the uncertainties in the values of the ERB fluxes become larger [...] "

418-421  "These gaps must be filled by the results of future experiments on satellites; for now, the extensive gaps in the experimental data for fluxes of ions with $Z \geq 2$ do not allow to create sufficiently complete and reliable empirical models of the ERB for these ions. "

599-603  "The numbers on the curves refer to the values of the decimal logarithms of J where J [...] is the differential flux of protons with [...] . Data of satellites are associated [...] symbols. The red lin corresponds to [...] , while green line corresponds to [...] "

605-609  See comments for lines 599-603

611-615  See comments for lines 599-603

617-621  See comments for lines 599-603

623-627  See comments for lines 599-603

629-633  See comments for lines 599-603

---

## Author Comment (AC1) · 27 Nov 2019

Reply to Interactive comment by Anonymous Referee #1 on the manuscript "Earth's radiation belts ions: patterns of the spatialenergy structure and its solar-cyclic variations" by Alexander S. Kovtyukh The manuscript addresses the scientific topic of understanding the spatial distributions of protons, Helium nuclei and CNO-group ions in the Earth Radiation Belts together with their variations during various solar minima and maxima. Even if it does not contain new data (it analyzes a wide range of older data from satellites), it offers a new interpretation of the mechanisms behind the different distributions of particle populations inside the magnetosphere. The exposition is fairly linear and the theoretical approach is sufficiently explained by the author, even if the description of the 3 sections seems rather repetitive, but it is something due to the very nature of the manuscript. The abstract reflects the content of the manuscript very well. Overall, it appears a little too long and with quite a few grammatical errors that make reading pretty hard and not fluent. Nevertheless, the topic is interesting, figures are simple but explanatory and the conclusions are satisfactory; thus, in my opinion, could be published after some fairly substantial grammar revision. I have a few more comments, then I will provide a table with some suggestions on how to correct some grammatical mistakes.

1. lines 96-97: what does "averaged" means here? And what does it mean that "All values of differential fluxes reduced to one dimension"? I think that this sentence should be rewritten more clearly. AC:These sentences reflect only the ordinary practice of working with experimental data. Apparently, they are surplus here and I removed them.

2. lines 144-147: this whole sentence, in my opinion, is not suitable for a scientific paper. The method used is the result, I believe, of your careful reasoning. Therefore the scientific community will evaluate and judge it on its own. You do not have to show any doubts about your choice. You can just address the fact that the uncertainties are linked to the errors of the experimental points shown in the various Figures. AC: I fully agree with you. The sentences removed from the text. A note about errors of the experimental points added.

3. lnes 148-157: in my opinion, describing in too much detail a procedure or a method that has not been used in the paper, does not help the linearity of the text and distracts the reader. This whole paragraph could be restricted to just two sentences. For example "[...] Representing plots in a different space of variables would lead only to more significant methodological errors and uncertainties, because of the natural differences in the instrumentation of the experiments taken into account; thus, a series of approximations or interpolation/extrapolation techniques would become inevitable.". AC: I agree with you. The sentences have been replaced by your sentence.

4. lines 179-180: I would remove this sentence because it does not add much to the description of the Figures. AC: I agree with you. The sentence iremoved from the text.

5. line 184: the function $f(\mu,K,L)$ is not previously cited nor defined. I guess it is a reference to line 30 and to the curves in the {E,L} plane, but maybe this implicit formula could be at least introduced earlier in the text. AC: I changed this sentence.

6. line 201: What is ? I can infer it is the slope of the flux (spectral index) but maybe in this occasion, it could be useful to, at least, describe (with a couple of words) what it means explicitly in these plots. AC: I made this addition.

Line Comment

1. I think that after ":" the P in "Pattern" should be lowercase AC: I agree. Text corrected.

6. In my opinion the word "ions" appears two times in too little space. I would write "[...] protons, helium and ions of [...] " AC: I agree. Text corrected.

8-9. "[...] considered here using data from satellites in the period 1961–2017" AC: I agree. Text corrected.

9-10 "It has been found that the results of these measurements line up [...] following some regular patterns" AC: I agree. Text corrected.

10-11 see line 6 AC: I agree. Text corrected.

11-14 This sentence is a little bit convoluted and it may result difficult for the reader to understand. I would suggest: "[...] It has been observed that in the inner regions of the ERB, ion fluxes decrease with increasing solar activity and that the solar-cyclic variations of fluxes of Z _ 2 ions are much greater than for protons; moreover, it seems that they increase with increasing atomic number Z." AC: I agree. Text corrected.

"Finally, the possible physical [...] " AC: I agree. Text corrected.

20-21 "The ERB consist mainly of [...] " AC: I agree. Text corrected.

23-24 The use of Z _ 2 here is redundant, in my opinion. Helium and Oxygen are already ions with Z _ 2 so I think that you can modify the sentence as "In ERB there are also helium nuclei and other Z >2 ions (like Oxygen etc) [...] " AC: I agree. Text corrected.

"[...] disturbances, ion fluxes, and their distributions are changed" AC: I agree. Text corrected.

I think that "[...] between a local vector [...] " should be "[...] between the local vector [...] " because in a well-determined region you can have only one specific vector of the magnetic field. Moreover, I would put a "the" also in front of "vector of a particle velocity" AC: I agree. Text corrected.

"which are injected [...] " AC: I agree. Text corrected.

"[...] drifted conserving [...] " AC: I agree. Text corrected.

"This layer is called the drift shell" AC: I agree. Text corrected.

32-33 I will suggest adjusting the following sentence as follows, to maintain the text more easily readable: "Therefore, experimental data on the ERB are often represented in coordinates {L, B}, where L is the drift shell parameter and B is the local induction of the magnetic field [...] " AC: I agree. Text corrected.

"center of the dipole itself (in Earth's radii RE)" AC: I agree. Text corrected.

In my opinion, the use of the passive form here should be avoided: "fluxes [...] decrease [...] " AC: I agree. Text corrected.

"[...] to higher latitudes [...] " AC: I agree. Text corrected.

I would add a "respectively" at the end of the sentence, to better distinguish between B and B0 AC: I agree. Text corrected.

42-43 "Outer and inner regions of the ERB are maintained in dynamic equilibrium with the environment by different mechanisms" AC: I agree. Text corrected.

44-46 I would suggest to slightly modify this sentence as follows: "The outer belt [...] is formed mainly by the mechanisms of radial diffusion of such ions towards the Earth under the action of fluctuations of both electric and magnetic fields resonating with their drift periods" AC: I agree. Text corrected.

46-48 I would avoid the repetition of the word "ions" here. Moreover, I would rewrite these parts: "This transport is accompanied [...] " and "as a result of their interactions [...] " AC: I agree. Text corrected.

49-53 I think that there is a temporal mismatch between "[...] protons with E > 10 MeV is formed [...] " and "The inner belt of ions with Z > 4 was formed [...] " AC: I agree. Text corrected.

54-55 "[...] the mechanism of ion capture from Solar Cosmic Rays takes place [...] " AC: I agree. Text corrected.

"[...] ERB, together with the sources of injection and losses of ions, are known" AC: I agree. Text corrected.

58-61 The sentence here is very important, in my opinion, because it introduces the problem that this paper tries to solve, so it should be written more clearly. "However, for comprehensive verification of the physical models and to identify the mathematical models and their parameters, the formulation of complete and reliable empirical models of the ERB for each of the ion components, is necessary; it is also necessary for ensuring the safety of space flights [...] " AC: I agree. Text corrected.

62-67 "These models can be created only using experimental data, obtained over many years and decades; such models [...] were already created for protons (AP8/AP9) and they are widely used in space research. On the contrary, measurements of Z _ 2 ion fluxes suffer from technical problems due to small statistics and high background of protons and electrons. For this reasons, empirical and semi-empirical models for Z _ 2

particles, are applicable [...] " AC: I agree. Text corrected.

68-72 I think that the first part of this sentence is just a repetition of what already said in lines 62-67; therefore I would leave only the part "One of the main problems of this work is to consider the possibility to create a sufficiently complete and reliable [...] " AC: I agree. Text corrected.

73-76 "In the following sections, the spatial-energy structure [...] (Sect. 2) together with the possible physical [...] are considered [...] , and the main [...] are given (Sect. 4) " AC: I agree. Text corrected.

78-79 The sentence here is a bit confused. I would suggest to change it as "There can be ions trapped in drift shells only with energies less than some maximum values, determined by [...] " AC:I agree. Text corrected.

gyroradius AC: I agree. Text corrected.

"[...] The green line in Figs 1-6 represents this very boundary [...] " AC: I agree. Text corrected.

87-88 I would remove the second "satellites " here and just leave the phrase as: "[...] with the differences in their trajectories [...] " AC: I agree. Text corrected.

"[...] sets of energy channels [...] " AC: I agree. Text corrected. 90 I would substitute "Solar" with "Sun" AC: I agree. Text corrected.

"[...] of the Earth during various periods of data-taking [...] " AC: I agree. Text corrected.

91-92 "[...] influence the fluxes of [...] with respect to proton fluxes [...] " AC: I agree. Text corrected.

94-95 "[...] In this section, experimental data of various [...] 90_ have been used." AC: I agree. Text corrected.

97-98 "[...] L shells, where these data were obtained, the ion fluxes are not distorted by the background [...] " AC: I agree. Text corrected.

99-103 "In many important experiments, the instruments were not able to separate fluxes of ions by their charge. Moreover, for the ions of the CNO group, the separation by mass are not usually performed. For heavier species, for example for Fe ions, we have very small data-sets. Therefore, this work presents data on helium ions (without any charge separation) and CNO ions (without any mass or charge separation)." AC: I agree. Text corrected.

104-108 "To solve the aforementioned problems, it is important to choose the form of representation (space of variables), in which the results of the single experiments can be compared to each other. In our case, the space {E. L} has been used; this choice is very efficient to better organize fragmentary experimental data obtained in different ranges of E and L." AC: I agree. Text corrected.

112-113 "Experimental points on these figures are connected by lines [...] " AC: I agree. Text corrected.

113-115 I think that a wide number of small sentences can break the fluency of the discussion. So, in this case, I would suggest to remove the last one and write: "[...] the decimal logarithms of the fluxes J, in unit of (cm2s ster MeV/n)−1, are shown near each iso-lines" AC: I agree. Text corrected.

"[...] but also very convenient [...] " AC: I agree. Text corrected.

118-119 I think that both sentences inside the parenthesis () could be removed here. AC: I agree. Text corrected.

120-143 This whole paragraph is crucial for the reader to better understand the meaning of Figures 1-6 and how they were built. So, the concept should be expressed in a more concise and linear way; therefore, I strongly suggest to review this paragraph. Meanwhile, I suggest a possible rewording: "The points in Figs. 1-6 have been obtained from the values of the single ion fluxes J(L) as a function of energy (the average energy for each channel of the instrument) and with an equatorial pitch angle close to 90_. Unlike electron fluxes or ion fluxes measured during geo-active conditions, the ion fluxes considered here (i.e. during quiet periods), usually have only one maximum. As a result, for each experiment, 1 or 2 points were obtained (both shown on the outer/inner edges of the E vs L profiles). Sometimes, especially for low fluxes, only one point was obtained: in these cases, the radial profile of the ion fluxes was cutoff at small values of L due to a significant background of contaminating particles and no interpolation/extrapolation has been performed whatsoever. Each iso-line, shown in these figures, has been evaluated separately from the corresponding set of experimental points (icons); thus, in more abundantly populated sectors of the plots (i.e. for protons with E > 1 MeV at L > 2) such iso-lines are mixing in Figs. 1-2. In case of a large distance between neighboring points, the corresponding segments of the iso-lines are shown as dashed arcs. The radial profiles of the differential fluxes J(L) of particles with different energy tend to intersect with each other in those regions where the energy spectra present some local maxima or minima. On the contrary, the iso-lines cannot intersect with each other: because this would mean that, at the same point in the {E, L} space, proton fluxes differ very significantly (by an order of magnitude). Such uncertainty does not have a physical sense and a special analysis is needed to identify other possible sources of error." I agree. Text corrected. 159-161 "[...] There is a large number of experimental data concerning ERB protons; the most important of them are presented in Figs. 1 and 2. These figures serve as a comparison with similar distributions of Z _ 2 ions (Figs. 3– 6)." AC: I agree. Text corrected.

"[...] Figure 1 represents results from the satellites [...] " AC: I agree. Text corrected.

"[...] have been collected during minimum periods of various solar cycles, i. e. between [...] " AC: I agree. Text corrected.

173-174 "[...] These results were obtained during maximum periods of solar cycles:

20th (1968-1971), 22nd (1990-1991), 20rd (2000), and 24th (2012- 2017) [...] " AC: I agree. Text corrected.

175-178 "The data [...] are given in both Fig 1 and 2 because solar-cyclic variations of the ERB proton fluxes are negligible [...] " AC: I agree. Text corrected.

181-184 "[...] the proton fluxes during solar minima (Fig. 1) are higher than during maxima (Fig. 2). In addition, in the former the inner edge of the proton belt is less steep and it can reach smaller L shells [...]" AC: I agree. Text corrected.

"[...] The red line corresponds to the [...] " AC: I agree. Text corrected.

192-194 "[...] can be connected to a discrepancy between the real configuration of the magnetic field lines of the magnetic field and the dipolar configuration (used here for L shell calculation) [...] " AC: I agree. Text corrected.

197-198 "[...] correspond to certain values [...] " AC: I agree. Text corrected.

"[...] changes of fluxes with changing L [...] " AC: I agree. Text corrected.

"[...] transformations in a magnetic field [...] " AC: I agree. Text corrected.

"[...] It results from these [...] " AC: I agree. Text corrected.

"distances [...] increase [...] " AC: I agree. Text corrected.

"In this work, a wider range of L and E is considered [...] " AC: I agree. Text corrected.

208-210 "[...] This is due to the fact that the energy range is significantly extended toward higher values (up to 200 MeV), but here the ionization losses for protons rapidly decrease [...] " AC: I agree. Text corrected.

I would put "are presented " at the end of this sentence AC: I agree. Text corrected.

"Figure 3 represents [...] " AC: I agree. Text corrected.

"Figure 4 represents [...] " AC: I agree. Text corrected.

215-224 For these sentences, I would suggest following the comments on lines 162-174 AC: I agree. Text corrected.

I suggest removing "are observed " I agree. Text corrected.

229-230 "[...AC: ] but the distribution of helium ions is slightly shifted towards higher values of L shell [...] " AC: I agree. Text corrected.

I think that could be useful to better define "white spots". Do you mean that there are few experiments on ERB helium studies? AC: I agree. Text corrected.

"AC: corresponds to the lower [...] " AC: I agree. Text corrected. 234 "[...] If one takes into account [...] " AC: I agree. Text corrected.

"[...] , which pass above the red line at L > 2.5, correspond to an average value of [...] " AC: I agree. Text corrected.

239-240 I would summarize a bit this sentence, writing only "[...] For helium ions spectra, as for protons ones, the values of the parameters of the power-law tail are in good agreement with what has been found in [...] " AC: I agree. Text corrected.

It is not clear to me what you intend here... I guess "[...] deviate from the slope of the red line [...] " -is that correct? AC: Text corrected.

244-245 "[...] increase with decreasing L slower than expected from adiabatic transformation. " AC: I agree. Text corrected.

245-247 "This means that the ionization losses of the ERB helium ions significantly exceed these losses for protons, in agreement to well-known calculations [...] " AC: I agree. Text corrected.

249-256 See comments on lines 162-174, 215 and 220 AC: Text corrected.

ËŸ[...] of the ion fluxes of the CNO group [...] " AC: Text corrected.

[Figure]

See my comments on line 230 AC: I agree. Text corrected.

"[...] but the fluxes of the CNO group increase by one order of magnitude or more [...] " AC: I agree. Text corrected.

263-264 "[...] it is seen also that the fluxes of the CNO group change several times more than the fluxes of helium ions do [...] " AC: I agree. Text corrected.

"[...] This means that [...] " AC: I agree. Text corrected.

269-270 "[...] to adiabatic laws that are not reported here, but this line let us estimate such deviations [...] " AC: Text corrected.

"[...] If one takes into account [...] " AC: I agree. Text corrected. 287 "[...] following the results [...] " AC: I agree. Text corrected.

"[...] of the ERB have been studied in many works [...] " AC: I agree. Text corrected.

291-292 "[...] These variations reach one order [...] " AC: I agree. Text corrected.

"[...] This density depends on [...] " AC: I agree. Text corrected.

"decrease " AC: I agree. Text corrected.

"[...] the stationary proton fluxes will increase with the decreasing solar activity [...] " AC: I agree. Text corrected.

307-308 "[...] and this leads to a decrease in the amplitude of the solar-cyclic variations of proton fluxes [...] " AC: I agree. Text corrected.

"[...] depends on its energy [...] " AC: I agree. Text corrected.

"[...] These variations can be explained by the same mechanism that has been suggested for protons at L < 2.5 [...] " IAC: agree. Text corrected.

315-316 "[...] For ions with Z _ 2 in the ERB, ionization losses are more significant than for protons and this can be connected to the absence of ion [...] " AC: I agree. Text corrected.

317-318 "[...] Such short lifetimes are manifested also [...] " AC: I agree. Text corrected.

"[...] $Z \_ 2$, the regions in which variations can manifest, should be [...] " AC: I agree. Text corrected.

"[...] in the energy ranges considered here [...] " AC: I agree. Text corrected.

"[...] of the same energy of the other ions under study [...] " AC: I agree. Text corrected.

"[...] These are very rough estimations, but they are in agreement with the results presented in [...] " AC: I agree. Text corrected.

329-330 "[...] for protons with $E > 10\text{-}20$ MeV at $L < 2.2$; in fact, protons form mainly under the action of the CRAND mechanism [...] " AC: I agree. Text corrected.

"form mainly [...] " AC: I agree. Text corrected.

333-335 "[...] the solar-cyclic variations of $Z \_ 2$ ion fluxes can be motivated only under the assumption that the effect related with an increase in the ionization losses of such ions significantly exceeds the effect connected with the possible enhance of radial diffusion of ions on the rising phase of solar activity [...] " AC: I agree. Text corrected.

"in logarithmic scale " AC: I agree. Text corrected.

340-341 "[...] the range of L, in which these dependencies for two energy channels are parallel to each other is connected to the power-law tail of the spectra [...] " AC: I agree. Text corrected.

341-343 "[...] Instead, on smaller values of L, these fluxes begin to converge and the radial dependencies of these fluxes intersect with each other, which is related to the maximum in the spectra. [...] " AC: I agree. Text corrected.

343-345 I suggest to remove this sentence here, or at least merge it with the following one on lien 345: "Concerning the physical mechanisms leading to the formation of power-law distributions of ions, the main source of ions in the outer [...] " AC: I agree. Text corrected.

"[...] usually spectra have an exponential shape [...] " AC: I agree. Text corrected.

"[...] The most likely region for this to happen is the plasma sheet [...] " AC: I agree. Text corrected.

"[...] the exponents of these spectra are close [...] " AC: I agree. Text corrected.

I think that the acronym for IMP-7 and 8 is well known to people reading this paper, I would leave only the reference to Sarris et al. AC: I agree. Text corrected.

"[...] the shape of the ion spectra of the PS usually do not change during [...] " AC: I agree. Text corrected.

359-360 "[...] in the PS exceed the times of substorms [...] " AC: I agree. Text corrected.

"[...] representations of the mechanisms [...] ". I would also suggest removing "and character" AC: I agree. Text corrected.

"[...] this part of the ion energy spectra is formed in the PS by stochastic mechanisms of ion acceleration; this hypothesis is supported by many experimental results [...] " AC: I agree. Text corrected.

367-372 "[...] The statistical aspect of these mechanisms reveals itself, in particular, in the fact that the ratios of fluxes (and partial densities) of ions with different Z can differ, even greatly, at low and high energies. During their passage in the phase space, ions gradually loose information about their origin and, therefore, the high-energy tails of their spectra contain ambiguous information on the partial densities of different components of ions in the source [...] " AC: I agree. Text corrected.

373-378 The high-energy portion of the ion spectra of the PS can be generated by the mechanisms of acceleration of particles on magnetic irregularities moving with respect to each other (Fermi mechanism). The fractal structures of the PS are revealed on scales from _ 0.4 to _ 8 thousands kilometers, for example, in the data of the satellite Geotail [...] If the mass of the ions are small compared to the mass of the magnetic irregularities in the PS, the average values of the index of the power-law tail should not depend on mass and charge of such nuclei " AC: I agree. Text corrected.

"it follows " AC: I agree. Text corrected.

"[...] <1, increases monotonically [...] " AC: I agree. Text corrected.

385-386 "[...] and their spectral density decreases rapidly with increasing [...] " AC: I agree. Text corrected.

"corresponds to the condition [...] " AC: I agree. Text corrected.

"[...] form analogous to that of ion spectra in the ERB [...] " AC: I agree. Text corrected.

398-399 "[...] In this work, the experimental results [...] plane, have been analyzed [...] " AC: I agree. Text corrected.

401-404 "[...] The degree of such similarity increases with [...] and it is linked to the nature of the main sources and on the universality [...] " AC: I agree. Text corrected.

405-407 "[...] Moreover, solar-cyclic (11-year) variations of the spatial-energy distributions of the ERB ion fluxes have been investigated, It has been noted that the ERB ions fluxes are weaker with increasing solar activity and this effect increases with increasing atomic number Z. " AC: I agree. Text corrected.

"[...] is typical, also, for faster [...] " AC: I agree. Text corrected. 409-410 "[...] as has been underlined in [...] " AC: I agree. Text corrected.

"[...] their radial diffusion can be neglected [...] " AC: I agree. Text corrected.

417-418 "[...] As Z and energy become larger and L becomes smaller, the uncertainties in the values of the ERB fluxes become larger [...] " AC: I agree. Text corrected.

418-421 "These gaps must be filled by the results of future experiments on satellites; for now, the extensive gaps in the experimental data for fluxes of ions with Z _ 2 do not allow to create sufficiently complete and reliable empirical models of the ERB for these ions. " AC: I agree. Text corrected.

599-603 "The numbers on the curves refer to the values of the decimal logarithms of J where J [...] is the differential flux of protons with [...] . Data of satellites are associated [...] symbols. The red line corresponds to [...] , while green line corresponds to [...] " AC: I agree. Text corrected.

605-609 See comments for lines 599-603 AC: I agree. Text corrected.

611-615 See comments for lines 599-603 AC: I agree. Text corrected.

617-621 See comments for lines 599-603 AC: I agree. Text corrected.

623-627 See comments for lines 599-603 AC: I agree. Text corrected.

629-633 See comments for lines 599-603 AC: I agree. Text corrected.

I am very grateful to Referee #1 for such an exclusively generous and thorough review. All these comments are very helpful for me and it is taken into account in the manuscript.

With grand regard, Alexander S. Kovtyukh

Please also note the supplement to this comment:
https://www.ann-geophys-discuss.net/angeo-2019-152/angeo-2019-152-AC1-supplement.pdf

———————————————

***Reply to Interactive comment*** by Anonymous Referee #1 on the manuscript "Earth's radiation belts ions: patterns of the spatial-energy structure and its solar-cyclic variations" *by* Alexander S. Kovtyukh

The manuscript addresses the scientific topic of understanding the spatial distributions of protons, Helium nuclei and CNO-group ions in the Earth Radiation Belts together with their variations during various solar minima and maxima. Even if it does not contain new data (it analyzes a wide range of older data from satellites), it offers a new interpretation of the mechanisms behind the different distributions of particle populations inside the magnetosphere. The exposition is fairly linear and the theoretical approach is sufficiently explained by the author, even if the description of the 3 sections seems rather repetitive, but it is something due to the very nature of the manuscript. The abstract reflects the content of the manuscript very well. Overall, it appears a little too long and with quite a few grammatical errors that make reading pretty hard and not fluent. Nevertheless, the topic is interesting, figures are simple but explanatory and the conclusions are satisfactory; thus, in my opinion, could be published after some fairly substantial grammar revision.

I have a few more comments, then I will provide a table with some suggestions on how to correct some grammatical mistakes.

1. lines 96-97: what does "averaged" means here? And what does it mean that "All values of differential fluxes reduced to one dimension"? I think that this sentence should be rewritten more clearly.

   These sentences reflect only the ordinary practice of working with experimental data. Apparently, they are surplus here and I removed them.

2. lines 144-147: this whole sentence, in my opinion, is not suitable for a scientific paper. The method used is the result, I believe, of your careful reasoning. Therefore the scientific community will evaluate and judge it on its own. You do not have to show any doubts about your choice. You can just address the fact that the uncertainties are linked to the errors of the experimental points shown in the various Figures.

   I fully agree with you. The sentences in lines 144-148 removed from the text. A note about errors of the experimental points added in lines 112-113.

3. lines 148-157: in my opinion, describing in too much detail a procedure or a method that has not been used in the paper, does not help the linearity of the text and distracts the reader. This whole paragraph could be restricted to just two sentences. For example "[...] Representing plots in a different space of

**Fig. 1.**

**Supplement:**

[revised manuscript text omitted]

---

## Referee Comment (RC2) · Anonymous Referee #1 · 29 Nov 2019

article [utf8]inputenc longtable

**Comments for the manuscript: "Earth's radiation belts ions: Patterns of the spatial-energy structure and its solar-cyclic variations" by A. S. Kovtyukh**

Thank you very much or the corrections! I think that the manuscript now is much linear

and easy to read. I have just a few more corrections though, mostly concerning the fluency of the text and no more technical questions. The list is reported below.

| Line | Comment |
|---|---|
| 12 | "[...] in the inner regions of the ERB, fluxes [...] |
| 24-25 | "The ERB consist mainly of electrons and protons, but there are also helium nuclei and other [...] " |
| 33 | "[...] geomagnetic trap, drift conserving [...] and populate [...] " |
| 34-35 | "This layer is called the drift shell. " |
| 37 | "For the dipole magnetic field, L is [...] " |
| 42 | "[...] along a certain magnetic field line [...] " |
| 43-44 | "This dependence is described [...] " |
| 45 | "[...] the same magnetic field line, respectively [...] " |
| 48-49 | "[...] of radial diffusion of ions towards [...] " |
| 53 | "The inner belt ($L < 2.5$) of protons with $E > 10$ MeV is formed by [...] " |
| 55 | "For protons with $E < 10$ MeV, this mechanism [...] " |
| 56 | "The inner belt of ions with $Z > 4$ is formed [...] " |
| 58-59 | "In the intermediate region ($2.5 < L < 3.5$), the mechanism of a ion capture from the Solar Cosmic Rays takes place during strong magnetic storms [...] " |
| 62 | "However, for a comprehensive verification [...] " |
| 76-77 | "[...] the possibility to create sufficiently complete and reliable empirical models [...] " |
| 79-83 | "In the following sections, the spatial-energy structure of the ERB in the {E, L} space for protons, helium and CNO group ions are considered (Sect. 2), together with possible physical mechanisms of formation of these structures and their solar-cyclic variations (Sect. 3). Finally, the main conclusions of this work are given (Sect. 4). " |
| 89 | "According to this criterion and to the theory of [...] " |

"[...] represents this very boundary [...] "

93-94 "A significant number of these discrepancies can be connected to the [...] "

107-108 "[...] to separate fluxes of ions by their charge. Moreover, for the ions [...] "

114-115 "[...] the results of every experiment can be compared to the others [...] "

"Figures 1–6 show the spatial-energy distributions [...] "

I suggest removing entirely the quote "hese figures united in pairs: " and just leave the part describing odd and even Figures

123-124 "The markers are connected by lines of equal intensity [...] "

I suggest removing the quote "In this place, it is need to say a few words about the method of constructing these figures.[...] "

"[...] corresponding set of experimental points (icons); then it was transferred [...] "

"Figure N sums up results from [...] "

188-189 "$21^{st}$ / $22^{nd}$ / $23^{rd}$ [...] "

See line 185

See lines 188-189

"From a comparison of Figs. 1 and 2, one can see [...] "

"[...] (2016a,b), which have been constructed from Figs. 1 and 2 confirm [...] "

"[...] J $\propto$ E$^{-\gamma}$, where the index $\gamma$ = [...] "

I would remove the "of the magnetic field" part here, magnetic field lines already describe everything

"Segments of iso-lines, that are parallel to the red line, also correspond to [...] "

"at L = 3–6, $\gamma$ = 4.8 $\pm$ 0.5. [...] "

"between these iso-lines increase with L [...] "

"[...] helium ion fluxes, averaged for quiet periods (Kp $<$ 2), are presented [...] "

See line 185

244-245 See lines 188-189

See line 185

See lines 188-189

"with Figs. 3–4, one can see that [...] "

253-254  "[...] E > 1 MeV practically do not change, and [...] "

"Figures 3 and 4 show the same patterns [...] "

"[...] because there are no experimental data for helium ions in these regions. [...] "

"For helium spectra [...] "

"[...] the red line (i.e. in the region of power-law spectra) substantially deviate from [...] "

"[...] CNO group ions fluxes, averaged for quiet periods (Kp < 2), are [...] "

See line 185

283-284  See lines 188-189

See line 185

286-287  "[...] period of activity [...] "

"[...] nd its configuration differ [...] "

"[...] Figs. 5–6 one can see that, for ions of CNO group, the [...] "

"[...] This means that, for ions of the CNO group, the ionization [...] "

"[...] have not been obtained by the experiments collected in [...] "

"[...] especially large at the peak of solar activity (Fig. 6): during these times, the slope of iso-lines [...] "

"At the same time, at L > 4 in Fig. 5 and at L > 3 in Fig. 6, the iso-lines [...] "

"[...] at the minimum of solar activity [...] "

"[...] following the results obtained [...] "

"[...] and are reduced rapidly with [...] "

"[...] have not been considered in these works [...] "

"In quiet periods, only the mechanism of ionization loss is significant [...] "

"[...] trapped in small L [...] "

"[...] the ERB protons are determined, in this mechanism, by the density [...] "

"[...] the proton supply rates to the inner belt, under the action of the CRAND mechanism, remain [...] "

"[...] with decreasing solar activity [...] "

"A proton lifetime on [...] "

"[...] this was noted in sections [...] "

"[...] where L* corresponds to the L shell of protons of the same energy [...] "

"[...] remains unchanged [...] "

363-364 "[...] in fact, these protons form mainly under the action [...] "

"[...] of radial diffuson of ions during the [...] "

"[...] values of L, these fluxes begin [...] "

"[...] highly turbulized region, but [...] "

"[...] must be generated in the outer [...] "

"The high-energy part of the ion [...] "

"[...] has a power-law shape and the exponents [...] "

394-395 "[...] along logarithmic axes E and J in a J(E) plane [...] "

"[...] decreases rapidly with [...] "

"Then, the lower boundary [...] "

"Using $B_s$ [...] "

434-435 "[...] as a result of their interactions with the current layer [...] "

"It has been found that in the outer belt [...] "

438-439 "[...] in the near equatorial plane [...] "

"[...] radial diffusion which conserves $\mu$ [...] "

"This kind of dependence of the amplitude [...] "

"[...] the extensive gaps in $Z \geq 2$ ion data do not allow [...] "

640-644 For what concerns all the Figures, the captions are very similar, so I suggest a modification in the first one (Figure 1) that should be repeated for all the others: "[...] J, which is given in units of (cm$^2$ s ster MeV)$^{-1}$, is the differential flux of protons [...] ". Also "associated with " and "[...] the power-law tail of the proton spectra, while green line corresponds to [...] "

---

## Author Comment (AC2) · 30 Nov 2019

Reply to Interactive comment by Anonymous Referee #1 from 29 November 2019 on the manuscript "Earth's radiation belts ions: patterns of the spatial-energy structure and its solar-cyclic variations" by Alexander S. Kovtyukh

Thank you very much or the corrections! I think that the manuscript now is much linear and easy to read. I have just a few more corrections though, mostly concerning the fluency of the text and no more technical questions. The list is reported below.

Line Comment 12 "[...] in the inner regions of the ERB, fluxes [...] AC: I agree. Text corrected.

24-25 "The ERB consist mainly of electrons and protons, but there are also helium nuclei and other [...] " AC: I agree. Text corrected.

"[...] geomagnetic trap, drift conserving [...] and populate [...] " AC: I agree. Text corrected.

34-35 "This layer is called the drift shell. " AC: I agree. Text corrected.

"For the dipole magnetic field, L is [...] " AC: I agree. Text corrected.

"[...] along a certain magnetic field line [...] " AC: I agree. Text corrected.

43-44 "This dependence is described [...] " AC: I agree. Text corrected.

"[...] the same magnetic field line, respectively [...] " AC: I agree. Text corrected.

48-49 "[...] of radial diffusion of ions towards [...] " AC: I agree. Text corrected.

"The inner belt (L < 2.5) of protons with E > 10 MeV is formed by [...] " AC: I agree. Text corrected.

"For protons with E < 10 MeV, this mechanism [...] " AC: I agree. Text corrected.

"The inner belt of ions with Z > 4 is formed [...] " AC: I agree. Text corrected.

58-59 "In the intermediate region (2.5 < L < 3.5), the mechanism of a ion capture from the Solar Cosmic Rays takes place during strong magnetic storms [...] " AC: I agree. Text corrected.

"However, for a comprehensive verification [...] " AC: I agree. Text corrected.

76-77 "[...] the possibility to create sufficiently complete and reliable empirical models [...]" AC: I agree. Text corrected.

79-83 "In the following sections, the spatial-energy structure of the ERB in the {E, L} space for protons, helium and CNO group ions are considered (Sect. 2), together with possible physical mechanisms of formation of these structures and their solar-cyclic variations (Sect. 3). Finally, the main conclusions of this work are given (Sect. 4). " AC: I agree. Text corrected.

"According to this criterion and to the theory of [...] " AC: I agree. Text corrected.

"[...] represents this very boundary [...] " AC: I agree. Text corrected.

93-94 "A significant number of these discrepancies can be connected to the [...] " AC: I agree. Text corrected.

107-108 "[...] to separate fluxes of ions by their charge. Moreover, for the ions [...] " AC: I agree. Text corrected.

114-115 "[...] the results of every experiment can be compared to the others [...] " AC: I agree. Text corrected.

"Figures 1–6 show the spatial-energy distributions [...] " AC: I agree. Text corrected.

I suggest removing entirely the quote "hese figures united in pairs: " and just leave the part describing odd and even Figures AC: I agree. Text corrected.

123-124 "The markers are connected by lines of equal intensity [...] " AC: I agree. Text corrected.

I suggest removing the quote "In this place, it is need to say a few words about the method of constructing these figures.[...] " AC: I agree. Text corrected.

"[...] corresponding set of experimental points (icons); then it was transferred [...]" AC: I agree. Text corrected.

"Figure N sums up results from [...] " AC: I agree. Text corrected.

188-189 "21st / 22nd / 23rd [...] " AC: I agree. Text corrected. (?)

See line 185 AC: I agree. Text corrected.

See lines 188-189 AC: I agree. Text corrected. (?)

"From a comparison of Figs. 1 and 2, one can see [...] " AC: I agree. Text corrected.

"[...] (2016a,b), which have been constructed from Figs. 1 and 2 confirm [...] " AC: I agree. Text corrected.

"[...] J / E−, where the index = [...] " AC: I agree. Text corrected.

I would remove the "of the magnetic field" part here, magnetic field lines already describe everything AC: I agree. Text corrected.

"Segments of iso-lines, that are parallel to the red line, also correspond to [...] " AC: I agree. Text corrected.

"at L = 3–6, = 4.8 ± 0.5. [...] " AC: I agree. Text corrected.

"between these iso-lines increase with L [...] " AC: I agree. Text corrected.

"[...] helium ion fluxes, averaged for quiet periods (Kp < 2), are presented [...] " AC: I agree. Text corrected.

See line 185 AC: I agree. Text corrected.

244-245 See lines 188-189 AC: I agree. Text corrected. (?)

See line 185 AC: I agree. Text corrected.

See lines 188-189 AC: I agree. Text corrected. (?)

"with Figs. 3–4, one can see that [...] " AC: I agree. Text corrected.

253-254 "[...] E > 1 MeV practically do not change, and [...] " AC: I agree. Text corrected.

"Figures 3 and 4 show the same patterns [...] " AC: I agree. Text corrected.

"[...] because there are no experimental data for helium ions in these regions. [...]"

AC: I agree. Text corrected.

"For helium spectra [...] " AC: I agree. Text corrected.

"[...] the red line (i.e. in the region of power-law spectra) substantially deviate from [...]" AC: I agree. Text corrected.

"[...] CNO group ions fluxes, averaged for quiet periods (Kp < 2), are [...] " AC: I agree. Text corrected.

See line 185 AC: I agree. Text corrected.

283-284 See lines 188-189 AC: I agree. Text corrected. (?)

See line 185 AC: I agree. Text corrected.

286-287 "[...] period of activity [...] " AC: I agree. Text corrected. (?)

"[...] nd its configuration differ [...] " AC: (?)

"[...] Figs. 5–6 one can see that, for ions of CNO group, the [...] " AC: I agree. Text corrected.

"[...] This means that, for ions of the CNO group, the ionization [...] " AC: I agree. Text corrected.

"[...] have not been obtained by the experiments collected in [...] " AC: I agree. Text corrected.

"[...] especially large at the peak of solar activity (Fig. 6): during these times, the slope of iso-lines [...] " AC: I agree. Text corrected.

"At the same time, at L > 4 in Fig. 5 and at L > 3 in Fig. 6, the iso-lines [...] " AC: I agree. Text corrected.

"[...] at the minimum of solar activity [...] " AC: I agree. Text corrected.

"[...] following the results obtained [...] " AC: I agree. Text corrected.

"[...] and are reduced rapidly with [...] " AC: I agree. Text corrected.

"[...] have not been considered in these works [...] " AC: I agree. Text corrected.

"In quiet periods, only the mechanism of ionization loss is significant [...] " AC: I agree. Text corrected.

"[...] trapped in small L [...] " AC: I agree. Text corrected.

"[...] the ERB protons are determined, in this mechanism, by the density [...] " AC: I agree. Text corrected.

"[...] the proton supply rates to the inner belt, under the action of the CRAND mechanism, remain ...] " AC: I agree. Text corrected.

"[...] with decreasing solar activity [...] " AC: I agree. Text corrected.

"A proton lifetime on [...] " AC: I agree. Text corrected.

"[...] this was noted in sections [...] " AC: I agree. Text corrected.

"[...] where L? corresponds to the L shell of protons of the same energy [...] " AC: I agree. Text corrected.

"[...] remains unchanged [...] " AC: I agree. Text corrected.

363-364 "[...] in fact, these protons form mainly under the action [...] " AC: I agree. Text corrected.

"[...] of radial diffuson of ions during the [...] " AC: I agree. Text corrected.

"[...] values of L, these fluxes begin [...] " AC: I agree. Text corrected.

"[...] highly turbulized region, but [...] " AC: I agree. Text corrected.

"[...] must be generated in the outer [...] " AC: I agree. Text corrected.

"The high-energy part of the ion [...] " AC: I agree. Text corrected.

"[...] has a power-law shape and the exponents [...] " AC: I agree. Text corrected.

394-395 "[...] along logarithmic axes E and J in a J(E) plane [...] " AC: I agree. Text corrected.

"[...] decreases rapidly with [...] " AC: I agree. Text corrected.

"Then, the lower boundary [...] " AC: I agree. Text corrected.

"Using Bs [...] " AC: I agree. Text corrected.

434-435 "[...] as a result of their interactions with the current layer [...] " AC: I agree. Text corrected.

438-439 "[...] in the near equatorial plane [...] " AC: I agree. Text corrected.

"It has been found that in the outer belt [...] " AC: I agree. Text corrected.

"[...] radial diffusion which conserves $\mu$ [...] " AC: I agree. Text corrected.

"This kind of dependence of the amplitude [...] " AC: I agree. Text corrected.

"[...] the extensive gaps in Z _ 2 ion data do not allow [...] " AC: I agree. Text corrected.

640-644 For what concerns all the Figures, the captions are very similar, so I suggest a modification in the first one (Figure 1) that should be repeated for all the others: "[...] J, which is given in units of (cm2 s ster MeV)$-1$, is the differential flux of protons [...] ". Also "associated with " and "[...] the power-law tail of the proton spectra, while green line corresponds to [...] " AC: I agree. Text corrected.

Deeply respected Referee #1, I am very grateful to you for such an exclusively generous and thorough review. All these comments are very helpful for me and it is taken into account in the manuscript.

With grand regard, Alexander S. Kovtyukh

[Figure]

**ANGEOD**

Please also note the supplement to this comment:
https://www.ann-geophys-discuss.net/angeo-2019-152/angeo-2019-152-AC2-supplement.pdf
* * *
[Figure]

**Supplement:**

[revised manuscript text omitted]
 are very grateful to the reviewers for their very important and fruitful comments and proposals for 
[revised manuscript text omitted]

---

## Referee Comment (RC3) · Anonymous Referee #1 · 2 Dec 2019

article [utf8]inputenc longtable

**Comments for the manuscript: "Earth's radiation belts ions: Patterns of the spatial-energy structure and its solar-cyclic variations" by A. S. Kovtyukh**

The manuscript has a very good overall structure now, thank you. My final list of

corrections is reported below. After that I think that the paper could be accepted without anymore interactions.

| | p12cm |
|---|---|
| Line | Comment |
| 49 | "The inner belt (L < 2.5) of protons with E > 10 MeV is formed mainly as a result of decay of neutrons knocked [...] |
| 49-50 | "In the intermediate region (2.5 < L < 3.5), the mechanism of a ion capture from the Solar Cosmic Rays takes place during strong magnetic storms " |
| 59 | I think that the word "models" is repeated too many times. I would substitute it here with "empirical representations" |
| 120 | "the ion fluxes considered here (i.e. during quiet periods), usually have only one maximum in the functions [...] |

---

## Author Comment (AC3) · 2 Dec 2019

Reply to Interactive comment by Anonymous Referee #1 from December 2, 2019 on the manuscript "Earth's radiation belts ions: patterns of the spatial-energy structure and its solar-cyclic variations" by Alexander S. Kovtyukh

The manuscript has a very good overall structure now, thank you. My final list of corrections is reported below. After that I think that the paper could be accepted without anymore interactions.

Line Comment

"The inner belt (L < 2.5) of protons with E > 10 MeV is formed mainly as a result of decay of neutrons knocked [...] AC: I agree. Text corrected.

49-50 "In the intermediate region (2.5 < L < 3.5), the mechanism of a ion capture from the Solar Cosmic Rays takes place during strong magnetic storms " AC: I agree. Text corrected.

I think that the word "models" is repeated too many times. I would substitute it here with "empirical representations" AC: I agree. Text corrected.

"the ion fluxes considered here (i.e. during quiet periods), usually have only one maximum in the functions [...] AC: I agree. Text corrected.

Deeply respected Referee #1, I am very grateful to you for such an exclusively generous and thorough review. All these comments are very helpful for me and it is taken into account in the manuscript.

With grand regard, Alexander S. Kovtyukh

Please also note the supplement to this comment:
https://www.ann-geophys-discuss.net/angeo-2019-152/angeo-2019-152-AC3-supplement.pdf
* * *
[Figure]

**Supplement:**

[revised manuscript text omitted]
 are very grateful to the reviewers for their very important and fruitful comments and proposals for 
[revised manuscript text omitted]

---

## Referee Comment (RC4) · Peter Kollmann (Referee) · 19 Dec 2019

This paper provides a compilation of radiation belt measurements from different missions from the 60s until the contemporary Van Allen Probes. This compilation is unique to my knowledge and a good reference for other studies.

Changes in intensity over solar cycle are studied in this paper. As different ion species are used, the observations can be compared with expectations of the mass dependence of different processes. Also, processes that are unique to protons can be ruled out. The observations are consistent with being shaped by energy loss in Earth's exosphere. This is a solid and useful scientific result. The study also supports that at

least protons are transported adiabatically, which is commonly assumed but less often properly proven.

A literally universal question in space physics is how particles are accelerated to the observed high energies. Analysis of the spectral shape in this paper suggests that MeV ions are accelerated through Fermi acceleration in the plasma sheet, which also is a valuable result.

I do have some comments, most importantly about the discussion and interpretation. Nevertheless, I see NO major problem here, so I suggest to accept the paper after MINOR revision. My detailed comments can be found in the attached PDF as highlighted text and sticky notes.

There are grammar issues in the article, even though less than in earlier papers. As I only highlighted a few of them, I suggest additional proofreading by a native speaker.

Best regards, Peter Kollmann

Please also note the supplement to this comment:
https://www.ann-geophys-discuss.net/angeo-2019-152/angeo-2019-152-RC4-supplement.pdf

**Supplement:**

[revised manuscript text omitted]

---

## Author Comment (AC4) · 29 Dec 2019

Reply to Interactive comment by Peter Kollmann (Referee #2) from December 19, 2019 on the manuscript "Earth's radiation belts ions: Patterns of the spatial-energy structure and its solar-cyclic variations" by Alexander S. Kovtyukh

Deeply respected Peter Kollmann, I am very grateful to you for such an exclusively

generous and thorough review. All these comments are very helpful for me and take into account in the manuscript.

I tried to take it into account as completely as possible. In my answers, the first line numbers refers to the original text of the manuscript, and the second numbers (after /) of the line refers to its final text.

I corrected grammar inaccuracies in the manuscript (thanks to Referee # 1). Corrections connected to your comments and with the supplement to your comments are given for the latest revised version of the manuscript (see RC1-RC3 and AC1-AC3 in the Interactive discussion of this manuscript).

Lines 9-10/9-11: I agree. Text corrected. Line 15/16-21: I agree. Text corrected. Line 22/28: I agree. Text corrected. Lines 30/35-37: I agree. Text corrected. Line 33/39: Here I do not quite understand what you mean. Lines 46-53/51-59: I add references. Line 83/86-87: I agree. Text corrected. I calculate isolines for smaller flux values. In some cases, break of the proton spectra in the region of the green line is observed. But this is unreliable: low-intensity proton fluxes are close to the instrumental background. Such work should be carried out for strong storms and the relaxation rates of the fluxes of captured protons should be considered depending on their energy. Line 98/98-99: I agree. Text corrected. Lines 125-126/123-124: I agree. I have make shorter and corrected this and subsequent paragraphs (until the end of the section). Lines 139-140/134-139: I agree. Text make shorter and corrected. I did not average the experimental data, but I used the averaged results in the corresponding papers. Line 145: This paragraph is destroyed as excess. As the errors of the experimental points in Fig. 1-6, I consider only standard deviation of the counts. Lines 159-161/145-147: I did not average the experimental data for a different satellite orbits, but I used the averaged results in the corresponding papers. Line 166: The results in Figs 1-2, as well as in Figs. 3-6, were received in missions that lasted for more than a year (usually 2-3 years or more). Line 191/176: The spectrum below the red line has a maximum, and between the maximum and the red line the spectrum is close to exponential. The

positions of the maximum and the exponential part change when L changes, according to adiabatic laws (in quiet periods). Below the maximum, the spectrum has a deep dip (during storms it is filled with ring current particles), and at the lower end of the dip, the spectrum is strongly indented (one or more peaks corresponding to auroral particles). I investigated all this in detail in 1984-2001 by other methods. In the figures given here, these spectral features are almost not visible, and I did not write about it. For other planets, see lines 394-398. Line 194/179: I agree. Text corrected. Lines 201/171-175: I agree. Text corrected. Line 214/197: Like many other things in the dynamics of the magnetosphere, this, of course, is a rather conventional upper boundary Kp for quiet conditions. Simply, all the data presented here were obtained at Kp <2 (see also lines 95/98 and 249/235). Lines 232-238/217-226; 276/262: Yes, I agree, with a small number of experimental data, this method does not work. The conclusions about the parameters of the power-law tail of the He and CNO ions were in fact bring here from my work (1984-2001), where they were obtained by other, more sophisticated and comprehensive methods, but they were all published in Russian journals. Text corrected. Lines 242-247/227-233: I agree. Text corrected. Lines 257-258/243-244: Yes, and it is also. Lines 268-270/254-256: I agree. Text corrected. Lines 294-296/281-298: I agree. Text corrected and supplemented. Lines 309-310/301-302 and 317-320: I agree. Text corrected. Lines 313/306-309: I agree. Text corrected. Lines 329/323-326: I agree. Text corrected. Lines 334-335/330-334: I agree. The text is supplemented. Line 377/373-375: I agree. Text corrected. Line 380/376: Yes, it is. Lines 382-383/380-381: I agree. Text supplemented. Lines 386-387/384-386: I agree. Text corrected and supplemented. Lines 389/387-388: I agree. Text supplemented. Lines 391/390-393: I agree. Text supplemented. Line 393/396: I agree. References have been supplemented. Line 401/403: I agree. Text corrected.

Deeply respected Peter Kollmann, I am very grateful to you for such an exclusively generous and thorough review.

With grand regard, Alexander S. Kovtyukh

Please also note the supplement to this comment:
https://www.ann-geophys-discuss.net/angeo-2019-152/angeo-2019-152-AC4-supplement.pdf

**Supplement:**

[revised manuscript text omitted]